# Endogenous stimulus-powered antibiotic release from nanoreactors for a combination therapy of bacterial infections

Yang Wu[1,4], Zhiyong Song [2,4], Huajuan Wang[3] & Heyou Han[1,2,3]*

The use of an endogenous stimulus instead of external trigger has an advantage for targeted and controlled release in drug delivery. Here, we report on cascade nanoreactors for bacterial toxin-triggered antibiotic release by wrapping calcium peroxide ($CaO_2$) and antibiotic in a eutectic mixture of two fatty acids and a liposome coating. When encountering pathogenic bacteria in vivo these nanoreactors capture the toxins, without compromising their structural integrity, and the toxins form pores. Water enters the nanoreactors through the pores to react with $CaO_2$ and produce hydrogen peroxide which decomposes to oxygen and drives antibiotic release. The bound toxins reduce the toxicity and also stimulate the body's immune response. This works to improve the therapeutic effect in bacterially infected mice. This strategy provides a Domino Effect approach for treating infections caused by bacteria that secrete pore-forming toxins.

[1] State Key Laboratory of Agricultural Microbiology, College of Life Science and Technology, Huazhong Agricultural University, No. 1 Shizishan Street, Hongshan District, Wuhan, Hubei 430070, China. [2] State Key Laboratory of Agricultural Microbiology, College of Science, Huazhong Agricultural University, No. 1 Shizishan Street, Hongshan District, Wuhan, Hubei 430070, China. [3] State Key Laboratory of Agricultural Microbiology, College of Food Science and Technology, Huazhong Agricultural University, No. 1 Shizishan Street, Hongshan District, Wuhan, Hubei 430070, China. [4] These authors contributed equally: Yang Wu, Zhiyong Song. *email: hyhan@mail.hzau.edu.cn

Multi-drug resistant bacterial infections have become one of the most pressing public health threats in the world[1,2]. Long-term and excessive use of antibiotic treatment will result in stronger resistance to bacteria and side effects on normal tissues, and several intensive efforts have been made in the area of advanced functional micro- and nanomaterials to avoid the side effects of current and developing therapies[3–5]. In this case, controlled drug release systems have been developed for the purpose of maintaining a therapeutically effective drug concentration in systemic circulation for a longer period of time, as well as reducing side effects by using an active substance at the right time and place, overwhelm drug resistance mechanisms with high, sustained local drug concentrations[6,7]. In this process, the concept of a nanoreactor was introduced for the design of a stimuli-responsive drug delivery and release nanosystem[8–11]. The potential applications of nanoreactors are not only involved in chemical synthesis, but also in many crosscutting fields such as biomedicine[12–14]. In particular, the in vivo use of micro-/nanoreactors has attracted the attention of more and more researchers for therapy and diagnosis of various diseases[15,16]. For construction of nanoreactors, the substrate and product should be exchanged between the inner and outer regions, that is, appropriate permeability is required for the wall of nanocompartments[17]. Moreover, the encapsulation of a wide variety of catalytic materials is another essential challenge. Despite the development of several nanoreactor systems, problems still remain in the encapsulation process and permeation of the substrate and products[18,19].

Pathogenic bacteria possess a range of virulence factors that enable them to colonize, invade, and replicate in immune competent hosts, and bacterial toxins are one of the most sophisticated virulence factors[20]. These effectors can target and disrupt cellular membranes, or act intracellularly and be highly specific to their target cells[21]. Alpha-toxin, also named α-toxin, is one of the major cytotoxic agents elaborated by Staphylococcus aureus (S. aureus) and the first bacterial exotoxin identified as a pore former[22]. These toxins disrupt cells by forming pores in cellular membranes and altering their permeability[23]. The pore size is ~ 2.5 nm, which promotes uncontrolled permeation of water, ions, and small molecules as well as rapid discharge of important molecules, such as ATP, dissipation of the membrane potential and ionic gradients, and irreversible osmotic swelling leading to cell lysis[22,24]. This strategy can be used for targeted treatment of bacterial infections to avoid ligand off-target problems and any damage to normal organizations.

In recent years, a new type of functional material, phase change material (PCM), has been found to be able to quickly respond to temperature and transform into a transparent liquid phase for a controllable release of drugs[25,26].

Thus motivated, here, we report an endogenous stimulus-powered targeted delivery and controllable drug release concept for the treatment of bacterial infections in combination with PCMs and toxin pore-formation activity (Fig. 1a). In this system, lecithin (Lec) and DSPE-PEG3400 are used to coat the eutectic mixture of two fatty acids as a gate material in fabricating toxin-responsive nanoreactors for drug release. Calcium peroxide ($CaO_2$) and rifampicin (RFP) are added into the eutectic mixture to form liposome-based nanoreactors. Once encountering pathogenic bacteria in vivo, the nanoreactors are pierced by the toxins secreted by the bacteria to form pores, and through the pores, water molecules enter the nanoreactors to react with $CaO_2$ and produce hydrogen peroxide ($H_2O_2$). Meanwhile, partial $H_2O_2$ decomposes to oxygen ($O_2$) to power the release of antibiotics, and the nanoreactors simultaneously stimulate the body's immune response after capturing bacterial toxins, significantly reduce the toxicity of

toxin and thus improve the therapy effect of bacterial infected mice.

## Results and discussion

### Design and characterization of liposome-based nanoreactors.
Our strategy for the rational design of an endogenous stimulus-driven liposome-based nanoreactor takes the advantage of the toxin that is secreted by the bacteria to form pores and in situ gas generation that drives the release of antimicrobial agents (Fig. 1b). As a natural saturated fatty acid, lauric acid (LA) and stearic acid (SA) have good chemical stability, biocompatibility, and degradability, thus are often used as a carrier for drug release[26,27]. It is similar to the previous report that the eutectic mixture formulated from LA (m.p. = 45.7–46.2 °C) and SA (m.p. = 71.8–72.3 °C) at a weight ratio of 4:1 exhibits a melting point at 35.2−38.3 °C[28], a temperature close to that of human bodies (36.2 −37.2 °C) (Fig. 1c and Supplementary Table 1), then the Lec and DSPE-PEG3400 were used to coat the eutectic mixture and form a toxin-reactive nanoreactor for drug release, which was mixed at a mass ratio of 3:1 to prevent hemolysis and also maintain the ability to adsorb toxin (Supplementary Fig. 1). The nanoreactors were characterized using the transmission electron microscopy (TEM) and the scanning electron microscopy (SEM). As shown in Fig. 1d and Supplementary Fig. 2, the image confirms the formation of a spherical structure with a relatively uniform size in the range of 150–200 nm. Based on the characteristic absorption spectra shown in Fig. 1e and ICP-MS (Supplementary Table 2), both RFP and $CaO_2$ were successfully loaded. When RFP-$CaO_2$@PCM@Lec (nanoreactors) are dissolved in ethanol, the absorption peak of RFP can be detected at 473 nm, but when dispersed in deionized (DI) water, the absorption peak cannot be detected, and then absorption peak can be recovered by adding toxin. $Ca^{2+}$ was successfully detected by ICP-MS. The drug loading contents were further determined to be $5.4 \pm 0.9\%$ and $17.2 \pm 1.2\%$ for RFP and $CaO_2$, respectively ($\pm$ SD, $n = 3$). Figure 1f shows the spectrum of LA-SA, with the peaks at 2917 cm$^{-1}$ and 2848 cm$^{-1}$ attributed to the stretching vibration of –$CH_3$ and –$CH_2$ group, respectively. The absorption peak at 1700 cm$^{-1}$ is assigned to the C = O stretching vibration. The peak at 721 cm$^{-1}$ corresponds to the out-of-plane bending vibration of the C–H group[29,30]. Meanwhile, the TEM elemental mappings showed the uniform distribution of C, N, O, P, and Ca elements, which also demonstrates the presence of $CaO_2$ and RFP in the nanoreactors, such as the Ca and N atoms shown in Fig. 1g.

### Stimulus-triggered drug release from the nanoreactors.
The ability of the nanoreactors to capture toxin was tested by mixing the toxin with different concentrations of nanoreactors, and 100 μg of the nanoreactors was found to be able to capture 4 μg of toxin (Fig. 2a and Supplementary Fig. 3). The immunoglod staining experiment showed that the nanoreactors without toxin treatment did not display any specific binding, whereas toxin-treated nanoreactor surface could combine very distinct gold nanoparticles. These results indicate that nanoreactors can efficiently capture toxins without affecting their structural integrity (Fig. 2b). The pore formation of nanoreactors was evaluated by SEM and fluorescence assays. SEM results indicated the formation of pores on the nanoreactors (Fig. 2c). The stability of liposomes was evaluated by using 8-aminonaphthalene-1,3,6-trisulfonic acid disodium salt (ANTS) and p-xylene-bis-pyridinium bromide (DPX) as a pair of fluorophore/quencher[21,31]. Figure 2d shows the fluorescence emission signals of ANTS/DPX-loaded nanoreactors in the presence of methicillin-resistant Staphylococcus aureus (MRSA) (secretes toxins), Bacillus subtilis (B. subtilis) (does not secrete toxins and is harmless to humans, plants

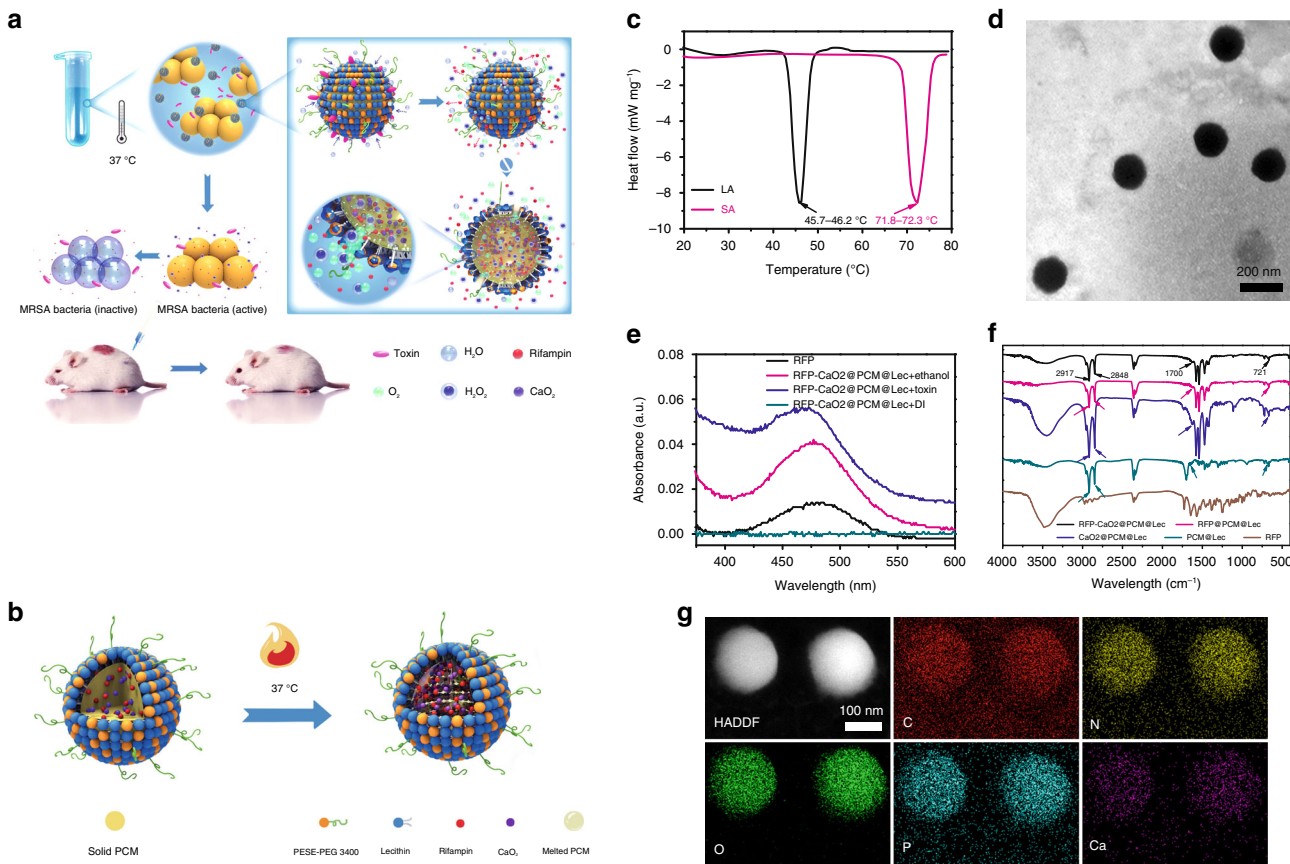

**Fig. 1** Design and characterization of liposome-based nanoreactors. **a** The scheme of endogenous stimulus-powered antibiotic release from RFP-CaO$_2$@PCM@Lec nanoreactors for bacterial infection combination therapy. **b** The solid PCM was dissolved in melted PCM at 37 °C. **c** Differential scanning calorimetry (DSC) curves of LA and SA. **d** A typical TEM image of the RFP-CaO$_2$@PCM@Lec nanoreactors. **e** UV absorption spectra of RFP under different conditions. Free RFP (black), RFP in ethanol (red), in toxin (blue) and in DI water (green). **f** FI-IR absorption spectra of different materials. **g** Mapping of RFP-CaO$_2$@PCM@Lec nanoreactors. Source data are provided as a Source Data file

and animals) and phosphate-buffered saline (PBS) buffer. It can be seen that a negligible signal was detected from ANTS when the nanoreactors were in contact with PBS buffer and *B. subtilis*; however, a significant signal increase occurred in the presence of pore-forming toxin secreted by MRSA bacterium. This result suggests that once the toxins insert into the membrane and form transmembrane pores, drug payloads can be released from the nanoreactors through these pores.

To confirm the above proposed mechanism, we prepared nanoreactors and evaluated the RFP drug release effect by incubating the as-prepared nanoreactors with the toxin at a different temperature (28, 30, 33, 35, 37 °C) for a different period of time (30, 60, 90, 120, 150 min). As shown in Fig. 2e, at a temperature below the eutectic point (35.2–38.3 °C), the nanoreactors exist in the solid state, thus preventing the leakage of payloads through diffusion. However, when the local temperature is above the eutectic point, the nanoreactors will melt, leading to a rapid release of payloads[32]. In our previous study[33], ORCA program[34] is employed to calculate the structure of RFP at the level of 6-311G(d, p), and the calculated data (Supplementary Fig. 4) showed that RFP has a diameter of 17.96 Å. The previous experimental and theoretical work indicates most atomic long-ranged interactions are >5 Å[35], thus RFP appears overly large to pass through even the largest pore. The CaO$_2$ encapsulated in the PCM nanoparticles reacts with the water through these pores, leading to the production of calcium hydroxide [Ca(OH)$_2$] (Fig. 2f) and hydrogen peroxide (H$_2$O$_2$) (Fig. 2g), with H$_2$O$_2$ spontaneously decomposed to form oxygen (O$_2$) (Fig. 2h). We

detected the content of H$_2$O$_2$ in the solution using the Hydrogen Peroxide Assay Kit. As shown in Fig. 2g and Supplementary Fig. 5, when the nanoreactors was incubated with the toxin at 37 °C, the yield of H$_2$O$_2$ in solution gradually increased within the 120 min time point, and the concentration of H$_2$O$_2$ reached a maximum of 2.09 mmol L$^{-1}$, accounting for 79.15% of the theoretical production. With the extension of time, the concentration of H$_2$O$_2$ was greatly reduced, probably due to the slow decomposition of H$_2$O$_2$ at 37 °C. However, when the nanoreactors was incubated with DI water at 37 °C, the maximum concentration of H$_2$O$_2$ was only 0.32 mmol L$^{-1}$ at the 60 min time point, which is only 12.10% of the theoretical production. These results further confirm that the toxin induces the formation of pores and then the water enters these pores to produce H$_2$O$_2$. As a medical reagent, H$_2$O$_2$ is widely used in wound disinfection to avoid bacterial infection and it is spontaneously decomposed to form oxygen[36]. As shown in Fig. 2h, a large amount of O$_2$ could be produced from the incubation of the nanoreactors with toxin, but not the DI water treatment, suggesting that the pore-forming toxin mainly contributes to the formation of pores and the water molecules can penetrate the pores and react with the CaO$_2$ to generate O$_2$. The creation of gas causes the volume of the nanoreactors to expand, and observable changes were found in the sizes of nanoreactors (Fig. 2i).

In order to confirm that the gas generation could drive the drug release, we prepared RFP@PCM@Lec in the absence of CaO$_2$ as a control. Supplementary Fig. 6 and Fig. 2j show the release profiles of RFP after incubation of RFP@PCM@Lec and

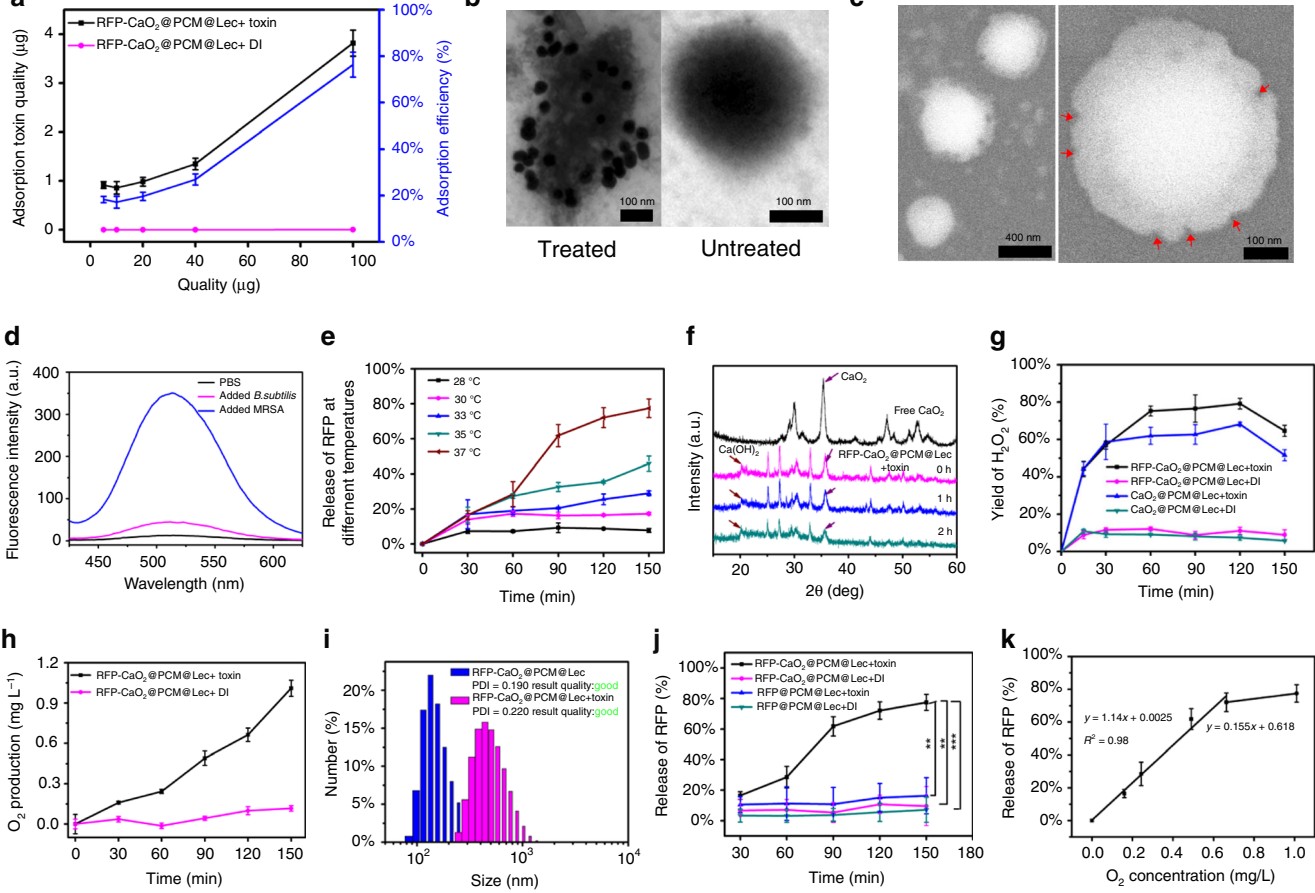

**Fig. 2** Endogenous stimulus-triggered drug release from the nanoreactors. **a** The RFP-CaO$_2$@PCM@Lec nanoreactors-captured toxin assay. **b** Transmission electron microscopy (TEM) images of nano-toxin immunostained with rabbit anti-toxin as the primary antibody and gold-labeled anti-rabbit IgG as the secondary antibody. **c** SEM images of toxin forming pore. **d** The fluorescence of ANTS recovered owing to the release of the ANTS under the influence of MRSA (blue), *B. sublitis* (red),and PBS buffer (black). **e** The RFP release from the RFP-CaO$_2$@PCM@Lec nanoreactors and RFP@PCM@Lec incubated with toxin and DI water at different temperatures (28, 30, 33, 35, 37 °C) and different periods of time (30, 60, 90, 120, 150 min). **f** Characterization of Ca(OH)$_2$ formation by XRD. **g** Ratio of H$_2$O$_2$ production to theoretical yield at different time points (15, 30, 60, 90, 120, 180, 360, 540, 720 min). **h** O$_2$ production at 37 °C at different time points (30, 60, 90, 120, 150 min). **i** The size changes after toxin was anchored into RFP-CaO$_2$@PCM@Lec nanoreactors. **j** The RFP release from the RFP-CaO$_2$@PCM@Lec nanoreactors and RFP@PCM@Lec incubated with toxin and DI water at 37 °C for different periods of time (30, 60, 90, 120, 150 min). **k** The correlation between gas generation and drug release. Error bars = standard deviation (*n* = 3). Source data are provided as a Source Data file

nanoreactors with DI water and toxin at 37 °C for a series of different time points (30, 60, 90, 120, 150 min). The results showed that the addition of CaO$_2$ greatly promoted the release of RFP, and the release of the drug was obviously improved with the prolongation of treatment time. Furthermore, we have evaluated the correlation between gas generation and drug release, and as shown in Fig. 2j, there was a significant positive correlation between gas production and drug release. The aforementioned results showed that the addition of CaO$_2$ could promote the release of antibiotics in a similar way to that of carbon dioxide (CO$_2$) power, i.e., the gas bubbles permeate the membrane and form a transient pore, through which to enable drug release[37–39]. Meanwhile, the Ca$^{2+}$ produced by the CaO$_2$ increases the concentration of ions inside the liposome, resulting in larger voids on the liposome and accelerating the escape of antibiotics in the manner of calcium phosphate as a drug delivery vehicle, so that drugs can successfully escape from lysosomes[40].

**In vitro target antibacterial activity of nanoreactors**. To prove that the nanoreactors can be targeted to combat against pathogenic bacteria, we selected MRSA as a model and *B. subtilis* as a control. As shown in Fig. 3a–c, the nanoreactors exhibit

antibacterial activity in a concentration-dependent manner, PCM@Lec has no obvious antibacterial activity against MRSA even at a high concentration (100 μg mL$^{-1}$), and 100 μg mL$^{-1}$ of nanoreactors displays a low antibacterial effect against *B. subtilis* (22.64%), but a significantly high antibacterial ability against MRSA (98.19%). However, when 100 μg mL$^{-1}$ nanoreactors and toxins are incubated together with *B. subtilis*, the inhibition rate has reached 96.71%, with the toxin treatment alone showing no effect on the bacterial growth (Supplementary Fig. 7). We also evaluated the antibacterial effect of 100 μg mL$^{-1}$ nanoreactors against MRSA. As shown in Fig. 3d, nanoreactors almost completely inhibited bacterial growth, but RFP@PCM@Lec, CaO$_2$@PCM@Lec, and PCM@Lec showed varying degrees of incomplete antibacterial effects, respectively. In vitro antibacterial activity (Fig. 3e, f) tests also showed that nanoreactors have efficient antibacterial ability (3.02 Log), and similar results were observed in live/dead staining (Fig. 3g). Furthermore, we evaluated the antibacterial efficiency of RFP and CaO$_2$ at different concentrations, and the pure RFP and H$_2$O$_2$ (Supplementary Fig. 8) were shown to have limited antibacterial activity. Overall, nanoreactors (96.71%) show higher antibacterial activity than RFP@PCM@Lec (30.87%) and CaO$_2$@PCM@Lec (40.85%).

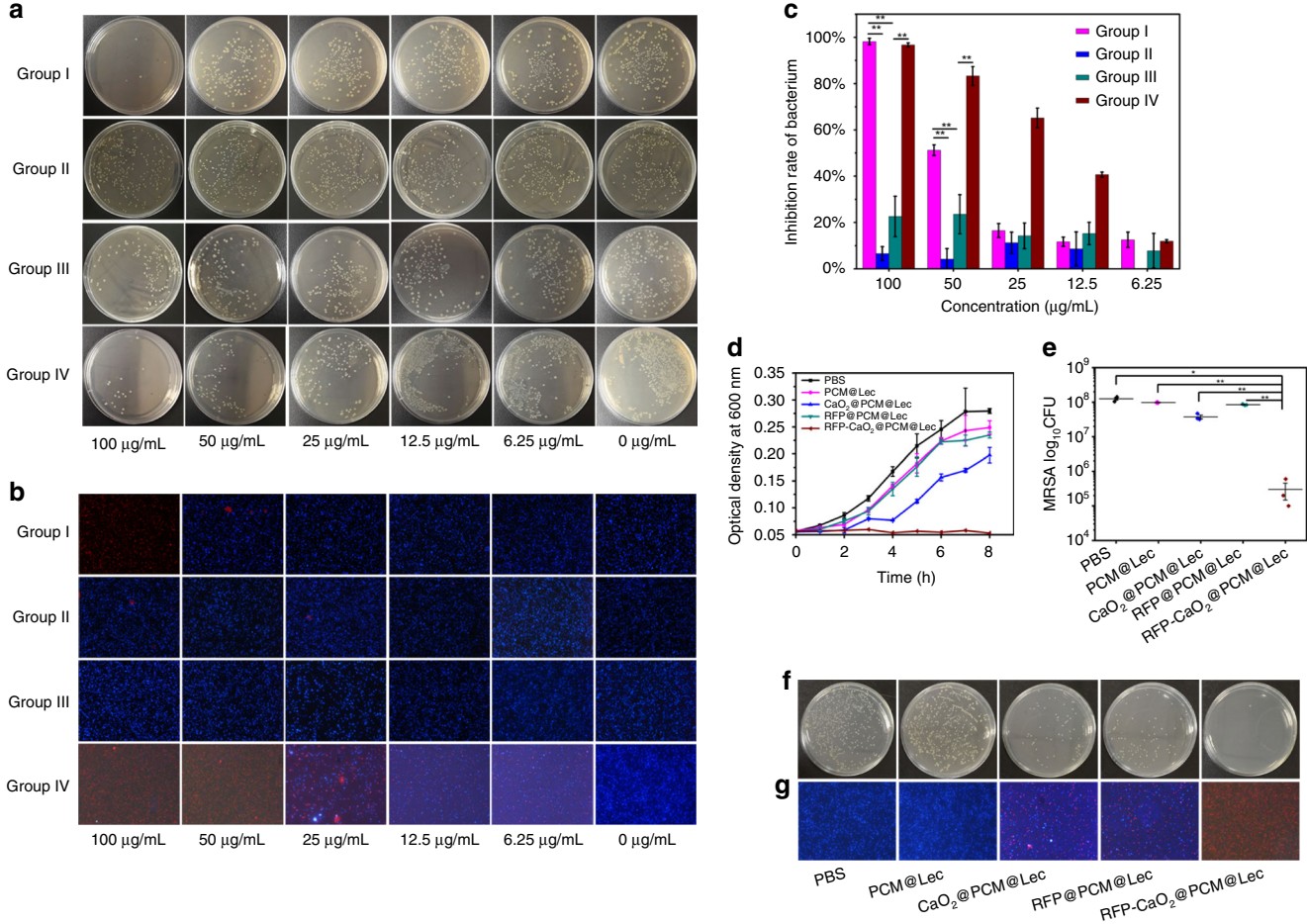

**Fig. 3** In vitro target antibacterial activity of nanoreactors. **a** Coated flat panel and **b** live/dead staining of MRSA and *B. subtilis* incubated with different concentrations of RFP-CaO$_2$@PCM@Lec nanoreactors (100, 50, 25, 12.5, 6.25, 0 μg mL$^{-1}$). group I RFP-CaO$_2$@PCM@Lec+MRSA, group II PCM@Lec +MRSA, group III RFP-CaO$_2$@PCM@Lec+*B.subtilis*, and group IV RFP-CaO$_2$@PCM@Lec + *B. subtilis* + toxin. **c** Bacterial inhibition rate of MRSA and *B. subtilis* incubated with different concentrations of RFP-CaO$_2$@PCM@Lec nanoreactors (100, 50, 25, 12.5, 6.25, 0 μg mL$^{-1}$). **d** Growth curve and **e** bacterial inhibition rate of MRSA incubated with 100 μg mL$^{-1}$ of different materials at 37 °C for 2 h. **f** Coated flat panel and **g** live/dead staining of MRSA incubated with 100 μg mL$^{-1}$ of different materials at 37 °C for 2 h. PBS, PCM@Lec CaO$_2$@PCM@Lec, RFP@PCM@Lec, and RFP-CaO$_2$@PCM@Lec nanoreactors are shown from left to right. The mean value was calculated by the *t* test (mean ± SD, *n* = 3). *$p < 0.05$, **$p < 0.01$, ***$p < 0.001$, compared with the indicated group. Source data are provided as a Source Data file

**Cytotoxicity and toxin-neutralizing ability of nanoreactors.** To verify the nanoreactors as a safe and effective approach for bacterial infection therapy, the cellular toxicity of nanoreactors was assessed using Vero cells. The relative cell viability was determined by standard MTT assay. After incubation with nanoreactors for 24 h, the relative cell viability was > 90% (Fig. 4a), indicating no obvious cytotoxicity on Vero cell proliferation was observed in the presence of 100 μg mL$^{-1}$ nanoreactors. Even when incubated at a concentration of 500 μg mL$^{-1}$, the relative cell viability was still > 90% (Fig. 4a). To evaluate whether the nanoreactors cause tissue damage, inflammation or lesion, histological analysis was performed in our study. As shown in Supplementary Fig. 9, the nanoreactor-treated mice group was normal, and there was no significant difference between the treated group and the control group. Blood biochemical analysis was performed to confirm the results of the histological analysis and to quantitatively evaluate the influence of nanoreactors on the exposed mice. No significant statistical difference was observed in the internal organs (heart, liver, spleen, lung, kidney) injured at a given dose of nanoreactors, which was further supported by the blood biochemical analysis (ALB, ALP, ALT, AST, A/G, BUN, GLOB, TP) (Fig. 4b). Based on the hematoxylin and

eosin (H&E) and blood biochemistry results, nanoreactors have no appreciable toxicity and are safe for in vivo application under our experimental conditions.

Furthermore, the toxin-neutralizing efficiency of nanoreactors was evaluated by measuring the hemolysis ratio of different nanoformulations (pure nanoreactors, free toxin, heat-inactivated toxin (heated toxin), and nanoreactors detaining toxin (nano-toxin)). As shown in Fig. 4c, d, the untreated free toxin has high hemolytic efficiency, but after the toxin is captured by the nanoreactor, the hemolytic rate decreases significantly. Moreover, the toxicity of different nanoformulations was assessed using the terminal deoxynucleotidyl transferase dUTP nick end labelling (TUNEL) assay and the results are shown in Fig. 4e and Supplementary Fig. 10. It can be seen that untreated free toxin caused a significant level of cellular apoptosis and inflammation, but when the toxin was captured by the nanoreactors, the level was significantly reduced.

The ability of the nanoreactors to neutralize α-toxin was further demonstrated in vivo by subcutaneous injection of pure nanoreactors, free toxin, heated toxin, and nano-toxin beneath the right flank skin of mice. Based on the skin lesions shown in Supplementary Fig. 11, the pure toxin induced demonstrable

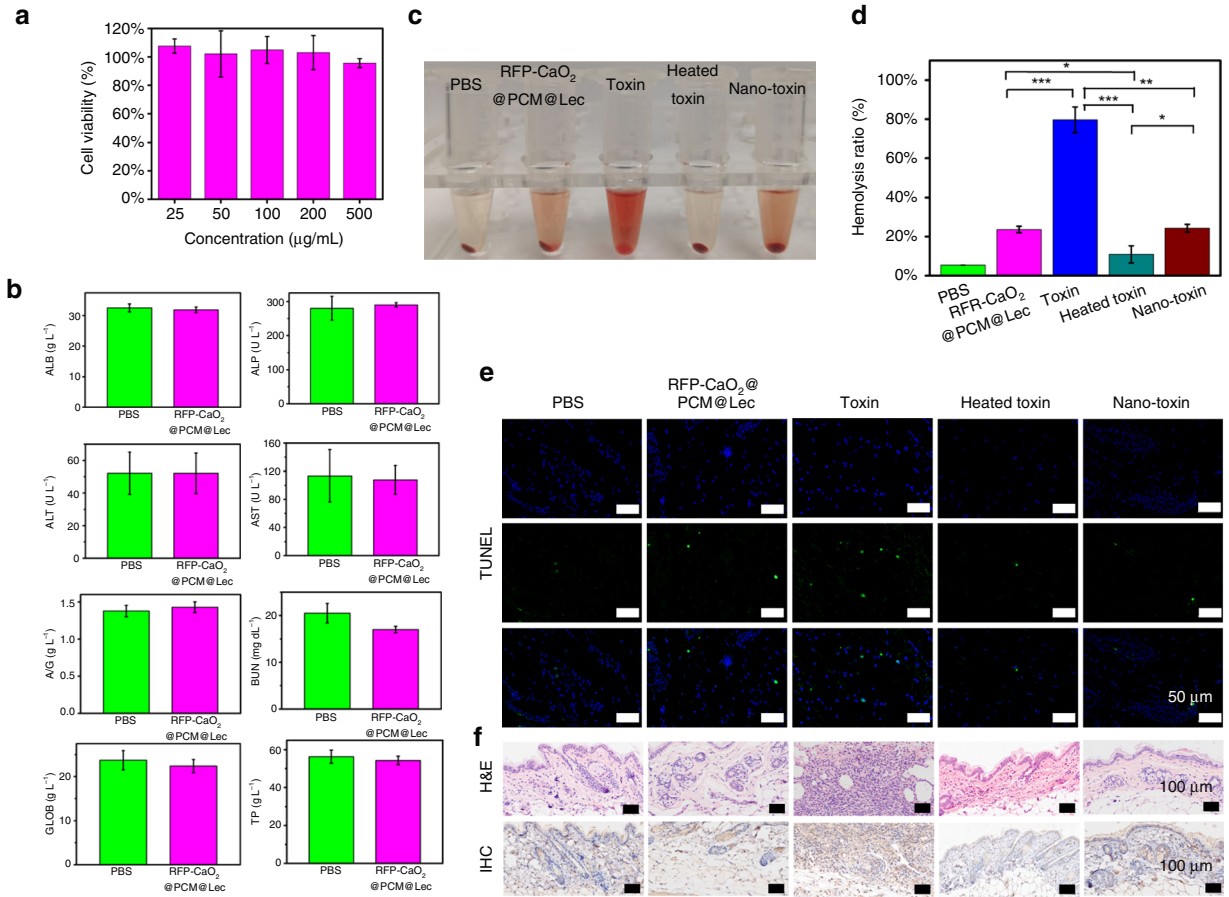

**Fig. 4** In vitro and in vivo safety evaluation. **a** MTT evaluation of RFP-CaO$_2$@PCM@Lec nanoreactors by using the Vero cell. **b** Blood biochemical analysis of RFP-CaO$_2$@PCM@Lec nanoreactors. **c** Representative images demonstrating the varying degrees of hemolysis. **d** Comparison of hemolysis induced by PBS, RFP-CaO$_2$@PCM@Lec nanoreactors, toxin, heated toxin, and nano-toxin ($n = 3$; mean ± SD). **e** TUNEL staining of skin samples collected from untreated mice or from mice at 48 h after subcutaneous injection of PCM@Lec, toxin, heated toxin, and RFP-CaO$_2$@PCM@Lec nanoreactors-captured toxin. **f** H&E and Immunocytochemistry (IHC) assays were used to evaluate the toxicity of different nanoformulations at 21 days post injection. The mean value was calculated by the $t$ test (mean ± SD). *$p < 0.05$, **$p < 0.01$, ***$p < 0.001$, compared with the indicated group. Source data are provided as a Source Data file

edema and inflammation with the extension of time (7 d, 14 d, 21 d), and this phenomenon became more and more serious, with obvious suppuration and muscle rot being observed in the skin tissue at the toxin injection site after 21 days of treatment. However, the nanoreactor-toxin showed no significant damage to the skin. Furthermore, the H&E, immunocytochemistry (IHC) and blood routine assays were used to evaluate the toxicity of different nanoformulations at 21 days post injection. The toxin treatment was shown to induce stronger tissue damage, inflammation or lesion by H&E and IHC analysis (Fig. 4f), in contrast to a similar result between the nanoreactor-toxin and the control, which was further supported by the analysis results of blood routine (Supplementary Fig. 12). All the above test results reveal that the nanoreactors can effectively neutralize toxins without causing significant cytotoxicity or physiological toxicity.

**In vivo antibacterial activity of nanoreactors**. The antibacterial capability of different treatments against MRSA infection was evaluated using a mouse skin infection model. MRSA represents one of the most common causes of skin infection both in the community and in hospitals[41]. The 1-cm diameter round wound was punched in the back of the BALB/c mice (6–8 weeks old) by using a skin punch. The mice were randomly divided into five groups based on five separate treatments with PBS buffer, PCM@Lec, CaO$_2$@PCM@Lec, RFP@PCM@Lec and

nanoreactors. As shown in Fig. 5a, each mouse was inoculated with 100 μL of 10$^6$ colony forming unit (CFU) per mL of MRSA by subcutaneous injection, followed by dropping 20 μL of 1 mg mL$^{-1}$ of different materials into the wound at 24 h post inoculation, and the treatment lasted 3 days. In order to evaluate the bactericidal effect, we excised the wound tissues and collected the internal organs (heart, liver, spleen, lung, kidney) and blood to quantify the number of bacteria at 4, 6, 8, and 10 days post treatment, with day 0 as a control. In Fig. 5b, it is shown that the wound area gradually decreased with increasing time, and the wound healing rate in the last group (nanoreactors) was significantly higher than that of the other groups. The wounds in the nanoreactor group had obviously healed by the 10th day, with their corresponding wound sizes (relative area vs. initial area) becoming much smaller each day than those of the other groups (Fig. 5c). From the grown colonies and the inhibitory efficiency (relative number of bacteria vs. initial number of bacteria) of the wound, we can see that the nanoreactor treatment exhibited better inhibition efficiency than the other groups (Fig. 5d, e). Meanwhile, we counted the number of bacteria in the organs and blood of mice every other day by the turbidity method (Supplementary Fig. 13), and the existence of nanoreactors was shown to effectively remove bacteria from various organs. The nanoreactor treatment significantly reduced the number of bacteria in the mice and showed better antibacterial effect than any other

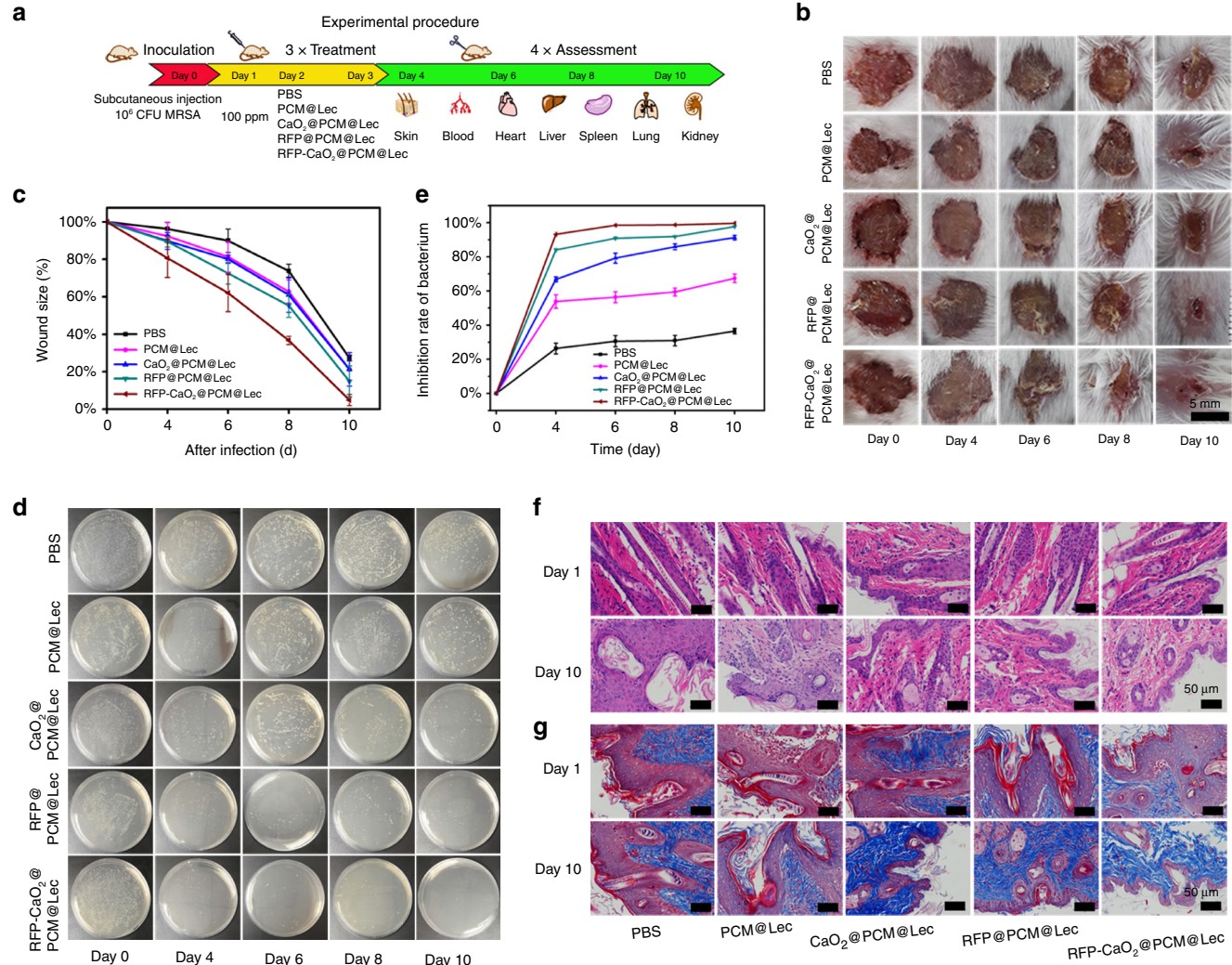

**Fig. 5** Evaluation of antibacterial activity in vivo. **a** The study protocol including MRSA inoculation and infection development on the BALB/c mice (6–8 weeks old), followed by the different treatments. **b** Photographs of wounds infected by MRSA. **c** Wound size (%) on different days (0, 4, 6, 8, 10). **d** The grown colonies and **e** the inhibitory efficiency (relative number of bacteria vs. initial number of bacteria) of wound. **f** H&E and **g** Masson images of the wounds on the first day and tenth day. Error bars = standard deviation ($n = 3$), Source data are provided as a Source Data file

treatment. Furthermore, H&E and Masson staining were performed on the tissues of mouse wounds on the first and the 10th day. As shown in Fig. 5f, g, all mouse wounds had a large number of inflammatory cells on the first day, whereas on the 10th day, only a small number of inflammatory cells were found on the wounds treated with nanoreactors. Masson's trichrome staining assay was used to verify the formation of collagen fiber (blue) during wound healing (Fig. 5g). Under the four control treatments, unrepaired collagen fibers were observed in the samples, whereas well-established collagen fibers were found in the samples under the nanoreactor treatment, which is consistent with the aforementioned test results.

**Immunity and in vivo toxin neutralization of nanoreactors.**
Following the in vivo antibacterial assessment, we studied the ability of the nanoreactors to elicit potent humoral immunity (Fig. 6a). Germinal centers are the primary sites for the affinity-based maturation of B cells, with the affinity of serum antibodies increasing with time after immunization[42,43]. These high-affinity antibodies of specific isotypes provide excellent protection against a variety of pathogenic microbial infections. To investigate the performance of different nanoformulations in immune effect and

in vivo toxin neutralization, draining lymph nodes were collected at 21 days post immunization to analyze the presence of B cells with the corresponding phenotype. Flow cytometric analysis showed that the toxin captured by the nanoreactors significantly increased the percentage of germinal center labeled GL-7 B cells to 24.43%, compared with 10.35% in the control group ($P = 0.04$) (Fig. 6b and Supplementary Fig. 14 and 15).

To test how the increased response to the nanoformulation is converted to antigen-specific immunity, we sampled the serum and analyzed the titer by indirect enzyme-linked immunosorbent assay (ELISA) at 21 days post immunization, around the peak of IgG responses (Fig. 6c). The result is consistent with a previous report on a nano-toxin formulated with purified toxin[44]. In the present study, the nano-toxin induced significantly higher toxin-specific antibody titers (100-fold, $p = 0.04$, $n = 6$) than the heat-treated toxin (60 min treatment). Furthermore, the in vivo toxin neutralization ability of nanoreactors was evaluated by measuring hemolysis ratio (Fig. 6d, e). It can be seen that the nanoreactors have better toxin-neutralizing ability and can significantly reduce the hemolysis rate. Finally, the protective immunity bestowed by the nanoreactors was evaluated by subjecting the vaccinated mice to toxin administration at a toxin dose of 120 µg kg$^{-1}$[45], which resulted in 100% mortality within 2 h in the unvaccinated group.

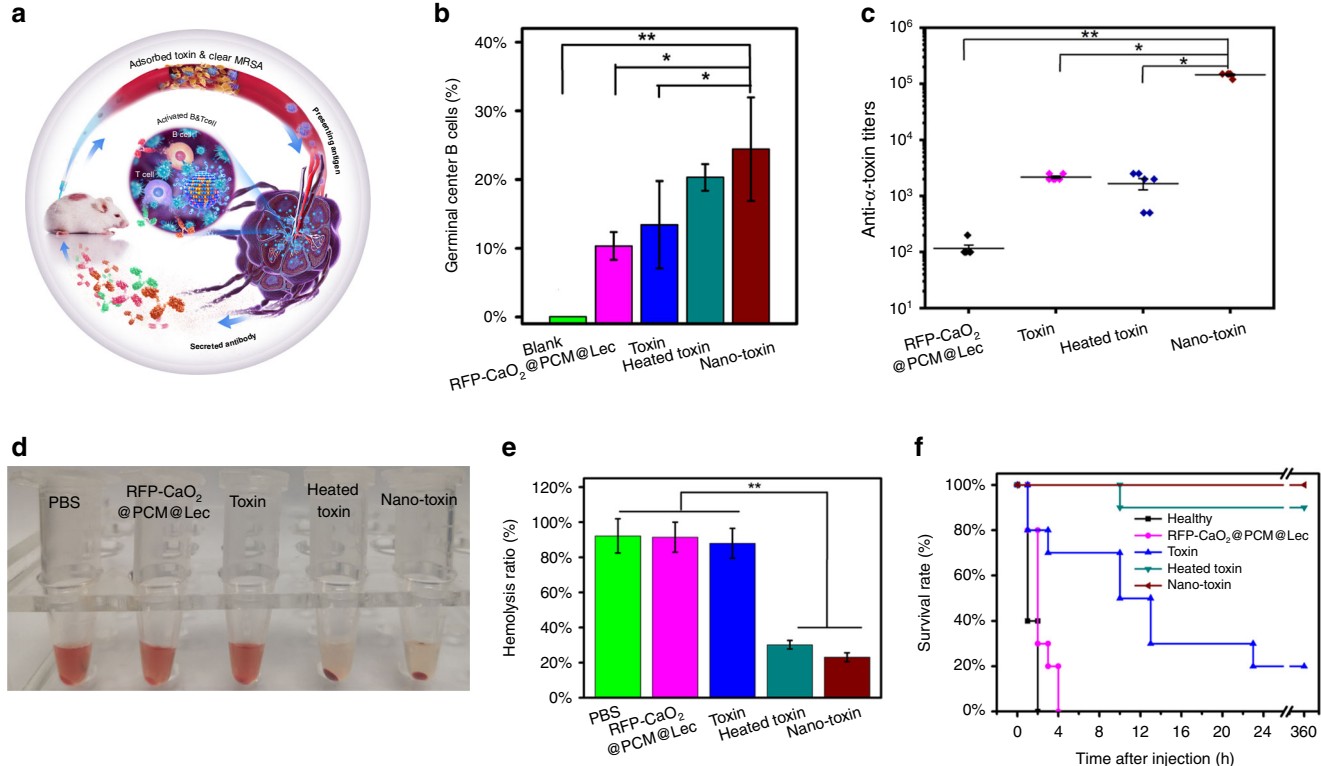

**Fig. 6** Immune effect and in vivo toxin neutralization of nanoreactors. **a** The scheme demonstrating that nanoreactors stimulate the body's immune response and improve the therapeutic effect of bacterially infected mice. **b** Flow cytometric analysis of cells in the draining lymph node at 21 days post administration with RFP-CaO₂@PCM@Lec nanoreactors, toxin, heated toxin, or nano-toxin ($n = 3$; mean ± SD). Cells were first gated on the B220 + IgD low population and values are expressed as percentage GL-7+. **c** Multivalent antibody responses in vivo. Mice were vaccinated with RFP-CaO₂@PCM@Lec nanoreactors, toxin, heated toxin, or nano-toxin on day 0 with boosts on day 7 and 14. On day 21 post first vaccination, the serum was sampled and analyzed for the presence of IgG antibody titers against toxin. **d** Representative images demonstrating the varying degrees of hemolysis. **e** Comparison of hemolysis induced by antibody generated by injection of PBS, RFP-CaO₂@PCM@Lec nanoreactors, toxin, heated toxin and nano-toxin ($n = 3$; mean ± SD). **f** Survival rates of mice over 15 days following an intravenous injection of α-toxin ($120 \,\mu g \, kg^{-1}$). The mean value was calculated by the $t$ test (mean ± SD). *$p < 0.05$, **$p < 0.01$, ***$p < 0.001$, compared with the indicated group. Source data are provided as a Source Data file

Meanwhile, the nano-toxin boosters improved the survival rate to 100% vs. an 80% survival rate for the heat-treated toxin vaccine with boosters ($n = 10$) (Fig. 6f).

In summary, a bacterial toxin targeted and oxygen-triggered antibiotic release system was developed based on a liposome-based nanoreactors. To ensure biocompatibility, the components used for fabrication of nanoreactors were obtained either from natural sources or from FDA-approved biocompatible polymers. The key mechanism was related to the phase change temperature and the gas-triggered sequential reaction. In this reaction, upon the contact of nanoreactors with the bacteria at 37 °C, the toxin could be anchored to the surface of nanoreactors and penetrate the layer of lecithin to form pores, followed by the entry of $H_2O$ molecules into the nanoreactors through the pores to react with the $CaO_2$ and produce $H_2O_2$. Meanwhile, part of $H_2O_2$ could decompose into $O_2$ to drive the controlled release of antibiotics. The structural integrity of the bacterial toxins captured by the nanoreactors was not destroyed, but on the contrary, the nanoreactor-toxin boosted the body's immune response to toxins, markedly reduced the toxicity of toxin and achieved a better therapeutic effect.

## Methods

**Fabrication of nanoreators**. In brief, LA and SA (4:1 by weight) were first dissolved in methanol at a concentration of $4 \, mg \, mL^{-1}$. Lecithin and DSPE-PEG3400 (3:1 by weight) were dissolved in 4% aqueous ethanol solution at a concentration of $1 \, mg \, mL^{-1}$. The phospholipid solution (15 mL) was heated to 50 °C. The PCM solution (3 mL) was mixed with the desired payloads (500 μL $5 \, mg \, mL^{-1}$ RFP in

dimethyl sulfoxide (DMSO) and/or 500 μL of $5 \, mg \, mL^{-1}$ $CaO_2$ in DMSO) and then added dropwise into the preheated phospholipid solution, followed by vigorous vortex for 2 min After cooling in ice water for 60 min, the cloudy solution was centrifuged for removing the un-encapsulated molecules and then filtered through a 0.22-micron filter. After washing three times with water, the resultant nanoreactors were suspended in water at 4 °C for further use[26].

**Synthesis of ANTS-DPX@PCM@Lec**. The phospholipid solution (15 mL) was heated to 50 °C. The PCM solution (3 mL) was mixed with the desired payloads (500 μL 12.5 mM of ANTS and 500 μL 45 mM of DPX in DMSO)[21,26] and then added dropwise into the preheated phospholipid solution, followed by vigorous vortex for 2 min After cooling in ice water for 60 min, the cloudy solution was centrifuged for removing the un-encapsulated molecules and then filtered through a 0.22-micron filter. After washing three times with water, the resultant nanoreactors were suspended in water at 4 °C for further use.

**Pore-forming assay**. First, we successfully synthesized the ANTS-DPX@PCM@Lec, while ensuring that there was no or weak fluorescence when dispersed in DI water. Then, the ANTS-DPX@PCM@Lec nanoparticles were treated with MRSA and *B. subtilis* at 37 °C for 2 h, with the PBS buffer used as a control. Finally, the resulting fluorescence emission intensity of ANTS in the filtrate was measured at 510 nm[21].

**O₂ production assay**. An appropriate amount of DI water was added to a well-sealed glass flask, then the probe of a portable dissolved oxygen analyzer (Lei-ci, Shanghai) was placed below the liquid surface at 37 °C. When the reading was stable, nanoreactors were quickly added into the device and were well sealed immediately. The readings were accurately recorded at different time points (0, 30, 60, 90, 120, 150 min).

**Evaluation of toxin adsorption and hemolysis of nanoreactors**. BCA Protein Assay Kit was used for quantitative detection of the adsorption of toxins by

materials. In brief, 200 μL of 500 μg mL$^{-1}$ nanoreactors synthesized in different mass proportions (Lec: DSPE-PEG = 1:1,3:1,6:1,9:1,12:1, and 1:0) was mixed with 10 μL of 400 μg mL$^{-1}$ toxin to interact with each other at 37 °C for 2 h, using PBS as a control. The mass of the adsorbed toxin was calculated by the absorbance at 462 nm according to the detection method of the BCA kit. Under the same experimental protocol, the hemolysis rate of the material can also be calculated by the following formula. In brief, 150 μL of different materials synthesized at different proportions (Lec: DSPE-PEG = 1:1, 3:1, 6:1, 9:1, 12:1, and 1:0) and 150 μL of 2% RBCs were incubated for 30 min at room temperature. After centrifugation at 2000 × $g$ for 5 min, the hemolysis was determined for each sample by measuring the absorbance of the supernatant at 540 nm using a microplate reader. A 100% lysis control was prepared by treating RBCs with Triton X-100. The hemolysis rate of each group was calculated as follows.

$$\text{Hemolysis rate} = \frac{\text{Abs(experiment)}}{\text{Abs(X} - 100)} \times 100\% \qquad (1)$$

**Bacterial culture**. In brief, 200 μL of 10$^8$ CFU mL$^{-1}$ bacteria was incubated with different concentrations of nanoreactors, RFP and CaO$_2$ at 37 °C for 2 h at 120 rpm. To evaluate the bacterial mortality, the treated bacteria were diluted and uniformly plated in Luria-Bertani (LB) solid medium, followed by incubation at 37 °C for 24 h. Finally, CFU was counted and compared with the control plate. Each treatment was prepared in triplicate and the mean values were compared with each other.

**In vivo safety**. In brief, the BALB/c mice (6–8 weeks old) were first shaved to remove the hair on the back. Subsequently, 200 μL of 100 μg mL$^{-1}$ of nanoreactors (20 μg) was injected subcutaneously, using PBS as a control. At 24 h post injection, the mice were killed, and the internal organs (heart, liver, spleen, lung, kidney) were collected for histological analysis by H&E staining. Meanwhile, the plasma was collected for biochemical indicator detection (ALB, ALP, ALT, AST, A/G, BUN, GLOB, TP).

Assessment was also performed on the toxicity of nanoreactors (100 μg), toxin (4 μg), heated toxin (4 μg, 70 °C inactivated for 1h), and nano-toxin (4 μg toxin absorbed by 100 μg RFP-CaO$_2$@PCM@Lec) using PBS as control. In brief, BALB/c mice were first shaved to remove the hair on their back and the above materials were injected subcutaneously and separately to each group of mice. At 24 h post injection, the mice were killed, and skin samples at the injection site were collected for histological analysis by H&E and TUNEL. TUNEL staining and Ipwin32 software were used to count the number of cells with a different color fluorescence.

After 21 days of immunization, H&E skin staining and IHC were performed on the dorsal skin of each group to judge the viable toxicity of different treatments. At the same time, the blood of the mice was collected, and blood routine tests were performed to observe the number of white blood cells and neutrophils (Gran). All animal experiments were in compliance with the Huazhong Agriculture University (HZAUMO-2018-036, approved by The Scientific Ethic Committee of Huazhong Agricultural University).

**Inhibitory effect of nanoformulations on hemolysis**. The ability of nano-formulations to prevent hemolysis was investigated under five different experimental groups: PBS, nanoreactors (100 μg), toxin (4 μg), heated toxin (4 μg, 70 °C inactivated for 1h), and nano-toxin (4 μg toxin absorbed by 100 μg nanoreactors). In brief, 150 μL of different materials and 150 μL of 2% red blood cells (RBCs) were incubated for 30 min at room temperature, followed by centrifugation at 2000 × $g$ for 5 min Next, the hemolysis of each group was determined by measuring the absorbance of the supernatant at 540 nm using a microplate reader. Meanwhile, a 100% lysis control was prepared by treating RBC with Triton X-100. Finally, the hemolysis rate of each group was calculated according to formula (1).

**Mice injury model**. The injury model was established on the BALB/c mice (6–8 weeks old). The 1-cm diameter round wound was punched in the back of the mouse using a skin punch, followed by infecting the wound with 100 μL of $1 \times 10^6$ CFU mL$^{-1}$ of MRSA, and the initial infection time was recorded as the zero day. After infection for 24 h, three mice were randomly killed for counting the bacteria of their wound, blood and internal organs by plate colony counting and turbidimetry. Furthermore, each wound was stained with H&E and Masson to identify whether the infection was established or not. The remaining mice were randomly divided into five groups, each group consisting of 12 mice. After the infection was established, the mice in different groups were treated separately with PBS buffer only, 20 μL of 1 mg mL$^{-1}$ PCM@Lec, CaO$_2$@PCM@Lec, RFP@PCM@Lec, and nanoreactors once a day for 3 days. After the wound was photographed, the mice were killed for counting the bacteria of their wound, blood, and internal organs by the same method at 4, 6, 8, and 10 days post infection. All animal experiments were in compliance with the Huazhong Agriculture University (HZAUMO-2018-036, approved by The Scientific Ethic Committee of Huazhong Agricultural University)

**Bacterial statistics**. Bacteria at the wound were counted using the plate colony counting method[43]. First, bacteria at the wound were collected by a cotton swab

and soaked in LB medium. After the medium was diluted 100 times, 20 microliters were applied for plate counting. Bacteria in internal organs (heart, liver, spleen, lung, kidney) and blood were counted using the turbidimetric method. First, the internal organs were crushed, then 20 μL of the polishing solution was taken out and spread in 180 μL of LB medium, using 20 μL of MRSA in logarithmic phase as a control. After incubation for 7 h, the optical density was measured at 600 nm by a microplate reader.

**Germinal center analysis**. BALB/c mice were first shaved to remove the hair on their back. The materials used included nanoreactors (100 μg), toxin (4 μg), heated toxin (4 μg, 70 °C inactivated for 1h), and nano-toxin (4 μg toxin absorbed by 100 μg nanoreactors), and PBS as control. These materials were injected subcutaneously on day 0, followed by a boost on day 7 and day 14. On day 21 after immunization, the lymph nodes were collected and dissociated into single cell suspensions for flow cytometric analysis. After staining the lymph nodes with APC Rat Anti-Mouse IgD (BD Pharmingen, 560868, 1:100), Alexa Fluor 488 Rat Anti-Mouse CD45R (BD Pharmingen, 557669, 1:100), PE Anti-mouse/human GL-7 Antigen (T- and B-cell Activation Marker) (BioLegend, 144607, 1:100), Purified Rat Anti-Mouse CD16/CD32 (Mouse BD Fc Block) (BD Pharmingen, 553142, 1:100) data were collected on a flow cytometer and analyzed using Flowjo software.

**Anti-α-toxin titer analysis**. Mice were subcutaneously administered with nanoreactors (100 μg), toxin (4 μg), heated toxin (4 μg, 70 °C inactivated for 1h) and nano-toxin (4 μg toxin absorbed by 100 μg nanoreactors) on day 0, followed by a boost on day 7 and day 14. On day 21, the serum of each mouse was collected for measuring toxin-specific antibody titers by an ELISA. First, a 96-well plate was coated overnight with 2 μg mL$^{-1}$ toxin using commercial coating buffer. Next, the wells were blocked with 1 wt% bovine serum albumin, followed by the addition of serially diluted serum samples as the primary antibody and horseradish peroxidase-conjugated Goat Anti-Mouse lgG (Sangon Biotech (Shanghai) Co., Ltd. D110087, 1:5000) as the secondary antibody. The plate was developed with H$_2$O$_2$/TMB-ELISA substrate, and the reaction was terminated by adding 2% sulfuric acid (H$_2$SO$_4$). Finally, toxin-specific antibody titers were measured at 450 nm using a Microplate reader.

**Toxin-neutralizing ability of nanoreactors**. After 21 days of immunization, sera were collected from the different experimental groups of mice. In brief, 20 μL serum was incubated with 10 μL of 50 μg mL$^{-1}$ toxin and 20 μL of Hank solution for 30 min at room temperature, followed by the addition of 50 μL of 2% RBCs and incubation for another 30 min According to the above experimental method, the hemolysis efficiency after serum toxin neutralization can be calculated by formula 1.

**Mouse survival rate**. After the end of the 21-day immunization, the mice treated with PBS, nanoreactors (100 μg), toxin (4 μg), heated toxin (4 μg, 70 °C inactivated for 1 h), and nano-toxin (4 μg toxin absorbed by 100 μg nanoreactors) were injected in the tail vein with toxin at the dosage of 120 μg kg$^{-1}$, and the survival rate of the decimals was observed for each group within 360 h.

**Statistical analysis**. All the results were presented as the mean value plus a standard deviation (± SD) from at least three independent experiments. Statistical analyses were performed using the $t$ test. Values of $*p < 0.05$, $**p < 0.01$, and $***p < 0.001$ were considered statistically significant.

## Data availability

The authors declare that data supporting the findings of this study are available within the paper and its supplementary information files. Source data are provided as a Source Data file by figshare (https://doi.org/10.6084/m9.figshare.9429287, hyperlink: https://figshare.com/s/ee97bba58d8aee5158c8).

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

## Acknowledgements

We are grateful to the financial support by the National Natural Science Foundation of China (21807036, 21778020), the Fundamental Research Funds for the Central Universities (2662016QD027), Sci-tech Innovation Foundation of Huazhong Agriculture University (2662017PY042, 2662018PY024), Science and Technology Major Project of Guangxi (Gui Ke AA18118046). We are also thankful to professor Hanchang Zhu for editing of the language, professor Xiangru Wang for providing us with the bacterial strain, Fengrui Wu for the help with SEM characterization.

## Author contributions

Y.W., Z.Y.S and H.J.W performed the experiments. Y.W. and Z.Y.S. were involved in data analysis. Y.W., Z.Y.S. and H.Y.H. designed experiments, interpreted results, and wrote the manuscript.

## Competing interests

The authors declare no competing interests.
