## [Peer Review File · Nature Communications]

Reviewers' comments:

Reviewer #1 (Remarks to the Author):

In the article "Endogenous stimulus-powered antibiotic release from nanomachines for a combination therapy of bacterial infections," the authors develop a liposome-based nanomachine capable of responding to the presence of alpha-toxin for bacterial treatment. The nanomachine consists of liposomes loaded with rifampicin and calcium oxide. Upon interaction with bacterial toxins, the nanomachine reportedly generates oxygen to accelerate drug release for antibacterial treatment. In addition, the authors highlight the nanocarrier's toxin-arresting capacity in triggering anti-toxin immune responses. A local *S. aureus* infection model was performed to highlight the treatment applicability of the platform. However, the treatment outcome is hardly impressive in my opinion. Overall, there are multiple glaring issues with the work that make it difficult for me to recommend its publication in Nature Communications. These issues are as follows:

1. The premise of the present work in terms of how the proposed nanoformulation may advance bacterial treatment is unclear. For example, antibiotics resistance is briefly mentioned, but how the controlled release system can address this issue is not described. Is rifampicin itself highly toxic and require controlled release? This premise should be defined and accompanied by comparison of the proposed formulation to the non-formulated drug in order to advance the field of antibiotics drug delivery. In addition, why rifampicin is chosen should be clearly defined.
2. The proposed nanoformulation hardly qualifies as "nanomachine" as claimed by the authors. Nanomachines typically are typically associated with mechanical movements, which are not described in the article at all.
3. The term, phase change material (PCM) is used repeatedly in the article, yet its significance is hardly discussed. Why is a phase change material necessary for the nanoformulation preparation? Can it not be prepared by simple liposomes?
4. The relative ease of the nanoformulation in binding pore-forming toxins is strange and unjustified given its PEGylated nature. Prior reports on nanoparticle-mediated toxin capture have all adopted non-pegylated formulations [JACS 2011, 133(11), 4132][Nature Biotechnology 33, 81-88 (2015)][Nature Nanotechnology 8, 336-340 (2013)], as the steric hindrance of PEG precludes protein interactions with nanoparticle surfaces. It is thus difficult to understand why the proposed PEGylated nanoformulation can interact with alpha toxin. Was there any optimization step to minimize PEG density to facilitate toxin interaction?
5. The authors report that the CaO can generate H₂O₂, which is in fact a very potent bactericidal agent. Its role in potentially improving antibiotics function should be examined and discussed.
6. Antibacterial activity should be expressed in logarithmic scale rather than linear scale. An effective antibiotic drug is typically associated with 3-log reduction in bacterial load.
7. H&E and Tunnel assays in figure 4d,e are hardly quantitative and don't depict how the absorbed toxin is safer than toxin.
8. The toxoid vaccine study feels out of place as it doesn't relate to the antibacterial study at all. The antibacterial study in Figure 7 was completed in 10 days and yet the antibodies were generated after 21 days. The value of the vaccination arm is therefore questionable in the present study.
9. The bacterial model with skin punch is strange and unjustified. Skin punch is typically applied in wound-healing studies. If the MRSA was injected via subcutaneous injection, why was the skin punch necessary? In fact, how could the bacteria be injected 'subcutaneously' as reported by the authors if the skin was removed at the site of examination?
10. The gating strategy is not described at all for the identification of germinal center B cell result in Figure 5b. The flow cytometric data should be provided.
11. Supplementary figures S5, S6, and S7 are not described at all in the text.

Reviewer #2 (Remarks to the Author):

How to solve multi-drug resistant bacterial infections is a serious problem at present. The current manuscript investigates the influence of the targeting of bacterial toxins by nanomachines coated with calcium peroxide and rifampicin and the mechanism by which gas-triggered nanomachines accelerate the release of antibiotics. This antibacterial strategy is absolutely new and quite interesting. Reviewer recommends the paper for publication in Nature Communications after addressing the following minor issues. Details are as following :

1. There are many small mistakes in the article. For example, in line 191 of page 8, "d to evaluate" should be "du". Please carefully check and correct them.
2. In Figure 1e, the different absorption intensity of RFP in the groups of Free RFP (black), RFP in ethanol (red), in toxin (blue) should be explained and the corresponding content of RFP should be provided.
3. In Figure 2j, why did the RFP@PCM@Lec+toxin cannot release RFP at 37 °C? Is the size of the RFP smaller than the pore induced by toxin? How about the structure of nanomachines at 37 °C for 150 min?
4. The clarity of the pictures in the article is not enough, the resolution of the pictures should be improved. Some pictures are wrong, for example, the plates in figure 3a are arranged differently (In Figure 3a, some images of group I and group II are placed repeatedly, please check out and correct it), and in figure 4, the descriptions of b and c are opposite.
5. The format of the references should be checked.
6. If possible, repeatability measurements (error bars) should be included in all episodes, and they need to put their overall impact results in the background. For example, there has been a significant reduction in the number of MRSAs after treatment, but they do not provide a reference to compare their results with current studies of current clinical MRSA treatments or new antibacterial treatments for MRSA.
7. How to prove that the calcium peroxide nanoparticles are coated inside the lecithin particles, rather than having some adsorption on their surface.
8. Obviously you should cite some of the latest antibacterial studies, such as the study of photothermal and photodynamic synergies.

Reviewer #3 (Remarks to the Author):

Comment:

In this manuscript, Han, et al. fabricate lipid (lauric acid and stearic acid) nanoparticles loaded with calcium peroxide (CaO₂) and rifampicin (RFM) as antibiotic and coated with liposome, comprising of lecithin and DSPE-PEG. The authors claim that this nanoparticle can absorb and neutralize pore-forming toxin, resulting in triggered release of antibiotic. Then this toxin absorbed nanoparticle can also elevate immune response against toxin. This work builds on two or more previously published works from Zhang's lab and Xia's lab. (shown in reference). The novelty is from using antibiotic triggered release in combination of vaccination effect of the same nanoparticles. Here, Han and colleagues present a good platform for combination treatment of bacterial infection and have done several experiments to proof their claims. However, there are some inconsistencies in the experiments and several important issues that are not discussed. As a result, this reviewer does not recommend this manuscript for acceptance in Nature Communication at this time.

Additional comments:

1. In results and discussion on evaluation of immune effect of nanomachines (line 209-221) and caption of figure 5 (line 403-409), approximately 80% of this paragraph and caption is exactly the same as previously published article. This considers as plagiarism and it is unacceptable, especially at Nature Communication's standard.

Wei X, Gao J, Wang F, Ying M, Angsantikul P, Kroll AV, Zhou J, Gao W, Lu W, Fang RH, Zhang L. In

situ capture of bacterial toxins for antivirulence vaccination. *Advanced Materials*. 2017 Sep;29(33):1701644.

2. In the manuscript line 91, nanoparticles size is 150-200 nm with relatively uniform size. However, SEM image in figure s1 and TEM image in figure 1b are quite poor quality and seems to be inconsistent. Particle size and polydispersity index (PDI) is commonly measured by dynamic light scattering (DLS). Size and PDI characterization by DLS should be provided.

3. What is drug loading and encapsulation efficiency of RFM and CaO₂ within RFP-CaO₂@PCM@Lec before and after 0.22 micron filtration?

4. What is concentration of RFM in RFP-CaO₂@PCM@Lec nanoparticles at 6.25, 12.5, 25, 50 and 100 µg/mL? The data should be provided and use free RFM as positive control for in vitro antibacterial activity study.

5. Why did you choose free fatty acid (lauric acid and stearic acid) at the ratio of 4:1?

6. Why did the author select rifampicin as antibiotic to treat MRSA infection? Please discuss why vancomycin which is the standard treatment for MRSA infection was not chosen.

7. In pore-forming assay, ANTS-DPX@PCM@Lec were synthesized. What are concentrations of ANTS and DPX used in the study and how is it fabricated?

8. In figure s3 (line 145-146), figure caption "The concentration of H₂O₂ generated by 1mg/mL RFP-CaO₂@PCM@Lec and RFP-CaO₂@PCM@Lec" is incorrect. This must be corrected.

9. Axis labels and texts in several figures throughout manuscript are quite small and low resolution.

10. Nanoparticle were called differently throughout manuscript such as RFP-CaO₂@PCM@Lec, nanomachines, nanotoxoid, nano-toxoid. It is very confusing.

11. If nanoparticles were claimed to neutralize toxins, hemolytic activity study should be provided.

12. In manuscript line 109-110, it is stated that pore forming toxin was investigated by SEM in figure 2c. It was incorrect. The result in figure 2c is TEM images. Moreover, if the pore size is 2.5 nm as mentioned in line 61, the results from TEM image in figure 2c indicates otherwise. It is inconsistent. Please discuss why did you think the arrow in figure 2c were pore-formed by toxin. Please show TEM images (negative staining) of untreated nanoparticle samples as negative control to compare with treated one.

Reference:

1. Hu CM, Fang RH, Luk BT, Zhang L. Nanoparticle-detained toxins for safe and effective vaccination. *Nature nanotechnology*. 2013 Dec;8(12):933.
2. Hu CM, Fang RH, Copp J, Luk BT, Zhang L. A biomimetic nanosponge that absorbs pore-forming toxins. *Nature nanotechnology*. 2013 May;8(5):336.
3. Wei X, Gao J, Wang F, Ying M, Angsantikul P, Kroll AV, Zhou J, Gao W, Lu W, Fang RH, Zhang L. In situ capture of bacterial toxins for antivirulence vaccination. *Advanced Materials*. 2017 Sep;29(33):1701644.
4. Zhu C, Huo D, Chen Q, Xue J, Shen S, Xia Y. A Eutectic Mixture of Natural Fatty Acids Can Serve as the Gating Material for Near-Infrared-Triggered Drug Release. *Advanced Materials*. 2017 Oct;29(40):1703702.

Point-by-point Response

Reviewer #1

Thanks for the positive comment and constructive suggestions on how to improve our manuscript.

Comments-1: *“The premise of the present work in terms of how the proposed nanoformulation may advance bacterial treatment is unclear. For example, antibiotics resistance is briefly mentioned, but how the controlled release system can address this issue is not described.”*

Response: Thank you for the good suggestion. In the newly manuscripts, we have added the detailed description about the how the proposed nanoformulation may advance bacterial treatment (**please see: P3, Lines 5-9**).

The misuse/overuse of antibiotics results in the generation of antibiotic-resistant bacteria, which would seriously threaten global public health (*Nature*, 2011, 476, 393). In this case, many triggering stimuli are explored for construction of controllable drug release systems (*Nat. Rev. Mater.*, 2016, 1, 16071; *J. Am. Chem. Soc.*, 2016, 138, 962; *Adv. Funct. Mater.*, 2018, 28, 1705137; *Adv. Funct. Mater.*, 2018, 28, 1800011), which can reducing side effects by improving antibiotic targeting and activity at the right time and place, which can overwhelm drug resistance mechanisms with high, sustained local drug concentrations (*Adv. Mater.*, 2012, 24, 6175; *Biomaterials*, 2016, 101, 207; *Angew. Chem. Int. Edit.*, 2016, 55, 8049).

In our study, we design and construct the cascade nanoreactors for bacterial toxin targeted and gas triggered antibiotic release by wrapping rifampicin and calcium peroxide (CaO₂). Our experimental results showed that the nanmoreactors can stimulate the body's immune response after capturing bacterial toxins and significantly reduce the toxicity of toxins, thereby improving the therapeutic effect of bacterially infected mice.

“Is rifampicin itself highly toxic and require controlled release? This premise should be defined and accompanied by comparison of the proposed formulation to the non-formulated drug in order to advance the field of antibiotics drug delivery.”

Response: Rifampicin (RFP) is one of the most potent and broad-spectrum antibiotics for treatment of tuberculosis (TB), leprosy and a growing number of Gram-positive bacteria such as multidrug-resistant *Staphylococcus aureus* (*Chemotherapy*, 1966, 11, 285). However, the 3-h half-life of RFP is believed to limit its utility for intermittent therapy, so new congeners or new strategies with long half-life are being developed (*Antimicrob. Agents Ch.*, 2007, 51, 3781).

Recently, developing novel drug formulations with sustained-release action, targeted delivery, high efficiency, and low toxicity has been suggested to help increase

the use of RFP to rapidly treat bacterial infection with high efficacy and reduce toxic side effect, maintaining a therapeutically effective drug concentration in systemic circulation for a longer period of time, thus potentially increasing patient compliance (*J. Cell Sci.*, 2013, 126, 3043; *ACS Appl. Mater. Interfaces*, 2014, 6, 16895; *Med. Sci. Monitor*, 2018, 24, 473).

“In addition, why rifampicin is chosen should be clearly defined.”

Response: In our study, RFP was chosen as the antibacterial drug against the Methicillin-resistant *Staphylococcus aureus* (MRSA) for the following reasons:

(1) RFP was primarily a frontline drug for the treatment of bacterial infection and showed good antibacterial activity in treating MRSA-related infections (*Proc. Am. Thorac. Soc.*, 2004, 1, 338; *Lancet*, 2001, 357, 40);

(2) In our previous research, RFP was loaded into the metal organic framework, and the combination of UV-light (365 nm), pH triggered precise RFP release and zinc ions enables the light-activated nanocomposite to significantly inhibit MRSA-induced wound infection and accelerate wound healing (*Adv. Funct. Mater.*, 2018, 28, 1800011);

(3) We used vancomycin, a common antibacterial agent for MRSA infection, as control, and the drug release efficiency in nanoreactors were evaluated. The results are shown in Figure 1, and it can be seen that the nanoreactors for RFP loading has better drug release ability.

As previously mentioned, we used RFP as the antibacterial agent in our study.

Figure 1 The release efficiency of nanoreactors for loading Vancomycin and Rifampin, respectively.

Comments-2: “*The proposed nanoformulation hardly qualifies as a nanomachine; as claimed by the authors. Nanomachines typically are typically associated with mechanical movements, which are not described in the article at all.*”

Response: Thank you for the good suggestion. Micro/nanoscale machines are a few micrometers to sub-micrometers scale devices that can harness power from various energy sources to generate mechanical motion in a controlled manner (*Adv. Funct. Mater.*, 2018, 28(25): 1705867).

In our study, Calcium peroxide (CaO₂) and rifampicin (RFP) are added into the lauric acid (LA) and stearic acid (SA) eutectic mixture to form phase change materials-based nanoformulations, then the DSPE-PEGylated-lecithin (Lec) is used to coat the nanoformulations as a gate material for fabricating toxin-responsive nanoparticles for drug release. Once encountering pathogenic bacteria in vivo, the nanoparticles are pierced by the toxins secreted by the bacteria to form pores, and a series of chemical reactions take place inside the nanoparticles to cause controlled release of the drug (**Please see: Equation 1 and 2**).

Therefore, we believe that the term nanoreactor is more suitable than nanomachine to illustrate our experiments based on these experimental phenomena and results. In the revised manuscript, we have replaced nanomachines with nanoreactors, and all the modified portions are highlighted in yellow.

Comments-3: “*The term, phase change material (PCM) is used repeatedly in the article, yet its significance is hardly discussed. Why is a phase change material necessary for the nanoformulation preparation? Can it not be prepared by simple liposomes?*”

Response: Thanks for the good suggestion. In the revised manuscripts, we have added the significance of the phase change material (PCM) (**please see: P4, Lines 9-11**).

In recent years, a new type of functional material, phase change material (PCM), has been found to be able to quickly respond to temperature and transform into a transparent liquid phase for a controllable release of drugs (*Angew. Chem. Int. Ed.*, 2014, 53, 3780; *Adv. Mater.*, 2017, 29, 1703702).

In our study, rifampicin (RFP) and calcium peroxide (CaO₂) were wrapped inside a phase-change material (PCM) which is made of a eutectic mixture of naturally occurring fatty acids with a well-defined melting point at 35.2°C-38.3°C (**Please see: Table S1 in the newly revised manuscript**), then the DSPE-PEG modified lecithin(Lec) is used to encapsulate the PCM to form a stable liposome, and because of the higher melting point of lecithin, the inner PCM could dissolve into a liquid state at temperature close to that of human bodies, and the melted PCM could be protected from leakage by the phospholipid layer, which can show we used the PCM rather than

the simple liposomes.

Our experimental results showed that the nanoparticles exist in the solid state thus preventing the payloads from leaking out through diffusion at a temperature below the eutectic point. However, when the local temperature is increased beyond the eutectic point, the nanoparticles will melt, leading to a quick release of payloads (**Please see: Figure S6 and Figure 2j in the newly revised manuscripts**).

Comments-4: *“The relative ease of the nanoformulation in binding pore-forming toxins is strange and unjustified given its PEGylated nature. Prior reports on nanoparticle-mediated toxin capture have all adopted non-pegylated formulations [JACS 2011, 133(11), 4132][Nature Biotechnology 33, 81-88 (2015)][Nature Nanotechnology 8, 336-340 (2013)], as the steric hindrance of PEG precludes protein interactions with nanoparticle surfaces. It is thus difficult to understand why the proposed PEGylated nanoformulation can interact with alpha toxin. Was there any optimization step to minimize PEG density to facilitate toxin interaction?”*

Response: Thank you for the good suggestion. In the newly revised manuscripts, we have supplemented relevant experiments to evaluate the effect of PEGylated on toxin adsorption. The results are shown in Figure 2 (**Please see: Figure S1 in the newly revised manuscript**), and it can be seen that PEGylated nanoformulations at different ratios (lecithin: DSPE-PEG, 1:1; 3:1; 6:1; 9:1; 12:1; 1:0) have a similar capture efficiency for alpha toxin, but the hemolysis assay showed that lack of PEGylated nanoformulation promotes hemolysis; meanwhile, as the PEG content increases, hemolysis ratio significantly decreases.

In this system, the optimized ratio of DSPE-PEGylated-lecithin (3:1) is used to coat the eutectic mixture of two fatty acids as a gate material for fabricating toxin-responsive nanoparticles for drug release.

Figure 2 The optimization step to minimize PEG density to facilitate toxin interaction. The capture efficiency of different-ratio nanoformulations for the toxin (a, b) and the hemolysis ratio of different-ratio nanoformulations (c and d).

Comments-5: “The authors report that the CaO₂ can generate H₂O₂, which is in fact a very potent bactericidal agent. Its role in potentially improving antibiotics function should be examined and discussed.”

Response: Recently, many researchers have studied reactive oxygen species [ROS; e.g., superoxide (O²⁻), hydrogen peroxide (H₂O₂), hydroxyl radical (•OH), and peroxynitrite (ONOO⁻)] for their versatile therapeutic applications such as wound healing (*Mater. Today*, 2017, 5, 238) and bacterial-infection treatment (*J. Am. Chem. Soc.*, 2016, 138, 3076). As a medical reagent, H₂O₂ is widely used in wound disinfection to avoid bacterial infection. Unfortunately, the limitations of H₂O₂ in antibacterial treatment mainly include low efficiency, slow process, and high concentration (*ACS Nano*, 2014, 8, 6202). In particular, traditional medical concentrations of H₂O₂ (0.5–3%) usually hamper wound healing and even damage normal tissues when treating bacterial infection (*Nanoscale*, 2018, 10, 17656).

In the newly revised manuscript, we have added the antibacterial assays using the RFP-CaO₂@PCM@Lec nanoreactors, RFP and CaO₂, respectively. The results are shown in Figure 3 (**Please see: Figure S8 in the newly revised manuscript**). It can be seen that the H₂O₂ from the CaO₂ can only inhibit 62% of bacteria; however, when RFP and CaO₂ were wrapped to form the nanoreactors, the antibacterial activity had a significant increase of ~100%.

Figure 3. Antibacterial efficiency of RFP-CaO₂@PCM@Lec nanoreactors (100 µg/mL), RFP (5.6 µg/mL) and CaO₂ (19.14 µg/mL).

Comments-6: “Antibacterial activity should be expressed in logarithmic scale rather than linear scale. An effective antibiotic drug is typically associated with 3-log reduction in bacterial load. Antibacterial activity should be expressed in logarithmic scale rather than linear scale. An effective antibiotic drug is typically associated with 3-log reduction in bacterial load.”

Response: Thanks for the good suggestion. In the newly revised manuscript, we have provided the logarithmic scale for antibacterial activity (please see: Figure 3e and 3f). The results are shown in Figure 4, and it can be seen that the RFP-CaO₂@PCM@Lec nanoreactors has 3.02 log reduction in bacterial load.

Figure 4. The antibacterial activity of different materials.

Comments-7: “H&E and Tunnel assays in figure 4d,e are hardly quantitative and don’t depict how the absorbed toxin is safer than toxin.”

Response: Thanks for the good suggestion. In this revised manuscript, we have added four assays to confirm that our synthesized nanoreactors have the ability to capture toxins, and the absorbed toxin is safer than toxin.

(1) Hemolysis ratio assay was used to evaluate the absorbed toxin and pure toxin. For detailed experimental methods, please see: P16, Lines6-20; The results have been provided in the newly revised manuscript (Please see: Figure 4c and 4d) and shown in Figure 5. It can be seen that the nano-toxin (absorbed toxin) has a lower hemolysis ratio than the pure toxin;

Figure 5. Hemolysis ratio assay of different materials. (a) Representative images demonstrating the varying degrees of hemolysis; (b) Comparison of hemolysis induced by PBS, RFP-CaO₂@PCM@Lec, toxin, heated toxin, and nano-toxin (n = 3; mean ± SD).

(2) Skin lesion assays were used to evaluate the demonstrable oedema and inflammation for different material treatments. The results have been provided in the newly revised manuscript (**Please see: Figure S11**) and shown in Figure 6. It can be seen that the nano-toxin (absorbed toxin) has weaker skin lesions than the pure toxin.

Figure 6. In vivo toxin neutralization. Mice injected with RFP-CaO₂@PCM@Lec, heated toxin, toxin, and nano-toxin. Skin lesions occurred at 7,14, and 21days following injection.

(3) Haematoxylin and eosin stained histological sections and blood routine examination assays were used to evaluate the inflammation with inflammatory infiltrate. For detailed experimental methods, **please see: P17, Lines 18-22**. The results have been provided in the newly revised manuscript (**Please see: Figure 4f and S12**) and shown in Figure 7. It can be seen that the nano-toxin (absorbed toxin) has weaker inflammation than the pure toxin (Figure 7a). Meanwhile, the blood routine examination assays showed that the pure toxin leads to higher white blood cell (WBC) and Granulocyte (Gran), and an increase in these indicators means that the

pure toxin induces stronger inflammation and body damage, while the nano-toxin did not produce significant inflammation and damage.

Figure 7. In vivo toxin neutralization. (a) Haematoxylin and eosin stained histological sections revealed inflammatory infiltrate, apoptosis, necrosis and oedema in the epidermis; (b) Blood Routine Examination.

(4) TUNEL staining assay was used to reveal the widespread apoptosis throughout. For detailed experimental methods, **please see: P 17, Lines 10-17**; the results have been provided in the newly revised manuscript (**Please see: Figure 4e and S10**) and shown in Figure 8. It can be seen that there was no obvious skin damage in the other four treatments, while the pure toxin displayed significant toxicity in vivo and induced stronger cell apoptosis.

Figure 8. TUNEL staining of skin samples collected from untreated mice or from mice at 24 h after subcutaneous injection of RFP-CaO₂@PCM@Lec, heated toxin, toxin, and nano-toxin. (scale bars = 50 μm)

Combining the above results, our experiments show that absorbed toxin is safer than toxin.

Comments-8: “The toxoid vaccine study feels out of place as it doesn’t relate to the antibacterial study at all. The antibacterial study in Figure 7 was completed in 10 days and yet the antibodies were generated after 21 days. The value of the vaccination arm is therefore questionable in the present study.”

Response: Thank you for the good suggestion. In the newly revised manuscript, we have added the *in vivo* detoxification assay to confirm that the nanoreactors injection can improve the survival rate of the toxin-challenged mice. For detailed experimental methods, **please see: P 20, Lines 12-16**; the results have been provided in the newly revised manuscript (**Please see: Figure 6f**) and shown in Figure 9. It can be seen that the nano-toxin vaccinations bestow strong protective immunity.

Figure 9. In vivo detoxification. Survival rates of mice over 15 days following an intravenous injection of α -toxin (120 μ g/kg)

At the same time, we provide the hemolysis induced by antibody generated from injection of PBS, RFP-CaO₂@PCM@Lec, toxin, heated toxin, and nano-toxin. For the detailed experimental methods (**Please see: P 20, Lines 6-11**); the results have been provided in the newly revised manuscript (**Please see: Figure 6d and 6e**) and shown in Figure 10. Our results showed that nano-toxin has the ability to induce stronger antibodies to neutralize the toxin.

Figure 10. Hemolysis ratio assay of antibody induced by different materials. (a) Representative images demonstrating the varying degrees of hemolysis; (b) Comparison of hemolysis induced by antibody generated from injection of PBS, RFP-CaO₂@PCM@Lec, toxin, heated toxin, and nano-toxin (n = 3; mean ± SD).

Comments-9: “The bacterial model with skin punch is strange and unjustified. Skin punch is typically applied in wound-healing studies. If the MRSA was injected via subcutaneous injection, why was the skin punch necessary? In fact, how could the bacteria be injected ‘subcutaneously’ as reported by the authors if the skin was removed at the site of examination?”

Response: *Staphylococcus aureus* skin infections represent a significant public health threat because of the emergence of antibiotic-resistant strains such as methicillin-resistant *S. aureus* (MRSA) (*Emerg. Infect. Dis.*, 2006, 12, 1715). Previous animal models to evaluate topical treatment of MRSA infections include a burned skin model (*Acta Biomater.*, 2015, 24, 87), a skin surgical/suture wound (*Antimicrob. Agents Chemother.*, 1976, 10(1): 38; *Antimicrob. Agents Chemother.*, 2006, 50, 3886), a slashed model (*Angew. Chem. Int. Edit.*, 2016, 55, 8049; *Nano Res.*, 2018, 11, 3762; *Adv. Funct. Mater.*, 2019, 29, 1808594) and skin punch model (*J. Invest. Dermatol.*, 2011, 131, 907; *J. Invest. Dermatol.*, 2018, 138, 1176).

In our study, the main reason for choosing the skin punch model is that we need to continuously monitor ex vivo bacterial burden using colony counts and repair of the wound under different treatment times (**For the detail methods, please refer to P 18, Lines 11-22**). The MRSA was injected via subcutaneous injection, and the main purpose of skin punch is to create a wound to facilitate bacterial and bacterial toxin enrichment.

Comments-10: “The gating strategy is not described at all for the identification of germinal center B cell result in Figure 5b. The flow cytometric data should be provided.”

Response: Thank you for the good suggestion. In the newly revised manuscript, we have provided the flow cytometric data in Figure 11 (**Please see: Figure S14 in the newly revised manuscript**).

Figure 11. The flow cytometric data of different treatments.

Comments-11: “Supplementary figures S5, S6, and S7 are not described at all in the text.”

Response: We feel very sorry for the mistakes in the old version of the manuscript. In the revised manuscript, we have added the discussion about the S5 (please see: P 9, Lines 7-9), S6 (please see: P12, Lines 8-11), S7 (please see: P 3, Lines 6-10, in the Supporting information).

Reviewer #2

Thanks for the positive comment and constructive suggestions on how to improve our manuscript.

Comments-1: “There are many small mistakes in the article. For example, in line 191 of page 8, “d to evaluate” should be “du”. Please carefully check and correct them.”

Response: We feel very sorry for the mistakes in the old version of the manuscript. In the revised manuscript, we have checked the manuscript carefully including the format and style, and all the modified portions are highlighted in yellow.

Comments-2: “In Figure 1e, the different absorption intensity of RFP in the groups of Free RFP (black), RFP in ethanol (red), in toxin (blue) should be explained and the corresponding content of RFP should be provided.”

Response: Thank you for the good comment and suggestion. In Figure 1e, the UV absorption spectra was only used to qualitatively evaluate the RFP, which showed that the RFP were successfully loaded with nanoreactors, but the ICP assay was used to quantitative analysis of RFP and CaO₂ loading rate(Please see: Table S2).

Comments-3: “In Figure 2j, why did the RFP@PCM@Lec+toxin cannot release RFP at 37 °C? Is the size of the RFP smaller than the pore induced by toxin?”

Response: In the newly revised manuscript, ORCA program (*WIREs. Comput. Mol. Sci.*, 2012, 2, 73) was employed to calculate the structure of RFP at the level of 6-311G (d, p), and the calculated data showed that the RFP has a diameter of 17.96 Å (please refer to the following Figure 1). The previous experimental and theoretical work indicates that most atomic long-ranged interactions are greater than 5 Å (*Phys. Rev. Lett.*, 2004, 92, 246401); however, the α -toxin pores are estimated to be 1–2.5 nm in diameter (*Biol. Cell*, 2006, 98, 667), thus RFP appears overly large to pass through even the largest pore.

At the same time, due to the lack of CaO₂ in the RFP@PCM@Lec nanoformulations, there is a lack of gas in the system as a driving force to promote the release of RFP, which further confirms the role of gas drive in the controlled release of drugs.

Figure 12. The structure of RFP at the level of 6-311G by using ORCA program. The distance between the two atoms at the edge of the blue wire frame is 17.96 Å. The red ball represents oxygen atom, white ball represents hydrogen atom, grey ball represents carbon atoms, and blue ball represents nitrogen atom.

“How about the structure of nanomachines at 37 °C for 150 min?”

Response: In the newly revised manuscript, we have supplemented relevant experiments and provided the representative images demonstrating that the change trend of nanoreactors (RFP-CaO₂@PCM@Lec) structure with the prolongation of toxin action time. The results in Figure 13 showed that the long-term treatment (24 h) leads to complete collapse of the nanocomposite structure; meanwhile, we have used the immunogold staining to confirm the presence of toxin protein.

Figure 13. Representative images demonstrating the change of the nanoreactors. (a-c) SEM images of toxin treatments at different times, and (d) TEM images of immunogold staining after 24-h toxin treatment.

Comments-4: *“The clarity of the pictures in the article is not enough, the resolution of the pictures should be improved. Some pictures are wrong, for example, the plates in figure 3a are arranged differently (In Figure 3a, some images of group I and group II are placed repeatedly, please check out and correct it), and in figure 4, the descriptions of b and c are opposite.”*

Response: We feel sorry to make this mistake in the old version. In this revision, we have checked the manuscript carefully, including each picture, and all the modified portions are highlighted in yellow.

Comments-5: *“The format of the references should be checked.”*

Response: Many thanks for the good suggestions. In this revision, we have checked the manuscript carefully, including each reference, and all the modified portions are highlighted in yellow.

Comments-6: *“If possible, repeatability measurements (error bars) should be included in all episodes, and they need to put their overall impact results in the background. For example, there has been a significant reduction in the number of MRSA after treatment, but they do not provide a reference to compare their results with current studies of current clinical MRSA treatments or new antibacterial treatments for MRSA.”*

Response: Many thanks for the good suggestions. In the newly revised manuscript, we have added the significant difference analysis in all episodes (**please see: Figure 2g, 3c, 3e, 4d, 6b, 6c, 6e**). All the modified portions are highlighted in yellow. At the same time, we have provided some reference to compare our results with current studies about the clinical MRSA treatments or new antibacterial treatments for MRSA (**please refer: Adv. Funct. Mater., 2018, 28, 1800011; Chem. Soc. Rev., 2019, 48, 415**), our results showed that the toxin stimulus-powered antibiotic release from nanoreactors have better antibacterial activity .

Comments-7: *“How to prove that the calcium peroxide nanoparticles are coated inside the lecithin particles, rather than having some adsorption on their surface.”*

Response: Thank you for the good suggestions. As we all know, the CaO_2 could react with water, leading to the production of calcium hydroxide $[\text{Ca}(\text{OH})_2]$ and hydrogen peroxide (H_2O_2). We detected the content of H_2O_2 in the solution using the Hydrogen Peroxide Assay Kit. The results showed that (**please refer to the Figure 2g in the newly revised manuscript**), when the RFP- CaO_2 @PCM@Lec was incubated with the toxin at 37°C , the yield of H_2O_2 in solution gradually increased within the 120 min time point, and the concentration of H_2O_2 reached a maximum of 2.09 mmol/L, accounting for 79.15% of the theoretical production. However, when the RFP- CaO_2 @PCM@Lec was incubated with DI water at 37°C , the maximum concentration of H_2O_2 was only 0.32 mmol/L at the 60 min time point, which is only 12.10% of the theoretical production. These results confirm that most of the CaO_2 was coated inside the lecithin particles.

Comments-8: *“Obviously you should cite some of the latest antibacterial studies, such as the study of photothermal and photodynamic synergies.”*

Response: Thanks for the recommendation of these relevant papers, which are important in the antibacterial research field. These corresponding studies are cited in the manuscript as the new Refs 5 (**please refer to P 3, Lines 5**).

Reviewer #3

Thanks for the positive comment and constructive suggestions on how to improve our manuscript.

Comments-1: *“In results and discussion on evaluation of immune effect of nanomachines (line 209-221) and caption of figure 5 (line 403-409), approximately 80% of this paragraph and caption is exactly the same as previously published article. This considers as plagiarism and it is unacceptable, especially at Nature Communication’s standard.*

Wei X, Gao J, Wang F, Ying M, Angsantikul P, Kroll AV, Zhou J, Gao W, Lu W, Fang RH, Zhang L. In situ capture of bacterial toxins for antivirulence vaccination. Advanced Materials. 2017 Sep;29(33):1701644.”

Response: We are very sorry about the high similarity with other articles. In the newly revised manuscript, we have rewritten the results and discussion sections about the evaluation of immune effect of nanoreactors (**Please see: P12, Lines 21-23; P 13, Lines 1-21**); meanwhile, the caption of figure 5 was also rewritten (**Please see: P 27, Lines 2-13**). In future articles, we must strictly abide by the academic norms and avoid the recurrence of similar problems.

Comments-2: *“In the manuscript line 91, nanoparticles size is 150-200 nm with relatively uniform size. However, SEM image in figure 1a and TEM image in figure 1b are quite poor quality and seems to be inconsistency. Particle size and polydispersity index (PDI) is commonly measured by dynamic light scattering (DLS). Size and PDI characterization by DLS should be provided.”*

Response: Thank you for the good suggestion. In the newly revised manuscript, we provided the high quality SEM and TEM image (**please see: Figure 1d and S2**). At the same time, we have provided the size and PDI characterization by DLS in our revised manuscript (**please see: Figure 2i**).

Comments-3: *“What is drug loading and encapsulation efficiency of RFP and CaO₂ within RFP-CaO₂@PCM@Lec before and after 0.22 micron filtration?”*

Response: Many thanks for the good suggestions. In the newly revised manuscript, we have provided the loading and encapsulation efficiency of RFP and CaO₂ within RFP-CaO₂@PCM@Lec before and after 0.22 micron filtration (**please refer to the Table 1**).

Table 1. The encapsulation efficiency of RFP and CaO₂ within RFP-CaO₂@PCM@Lec before and after filtration

Materials	Loading rate of RFP (%)		Loading rate of CaO ₂ (%)	
	Before	After filtering	Before	After filtering

	filtering		filtering	
RFP-CaO ₂ @PCM@Lec	8.5% ± 0.1%	5.4% ± 0.9% (*)	20.6% ± 3.3%	17.2% ± 1.2%
RFP@PCM@Lec	10.9% ± 4.0%	8.2% ± 1.1%	0	0
CaO ₂ @PCM@Lec	0	0	28.2% ± 3.1%	21.9% ± 1.8% (*)
PCM@Lec	0	0	0	0

Comments-4: “What is concentration of RFP in RFP-CaO₂@PCM@Lec nanoparticles at 6.25, 12.5, 25, 50 and 100 µg/mL? The data should be provided and use free RFP as positive control for in vitro antibacterial activity study.”

Response: Thank you for the good suggestions. In the newly revised manuscript, we have provided the antibacterial activity about different concentrations of RFP-CaO₂@PCM@Lec nanoparticles, using free RFP as positive control, and the results shown in Figure 14 (please see: Figure S8) suggest that the pure RFP and H₂O₂ have limited antibacterial activity compared with the RFP-CaO₂@PCM@Lec nanoparticles.

Figure 14. Antibacterial activity of RFP-CaO₂@PCM@Lec nanoreactors (100, 50, 25, 12.5, 6.25 µg/mL), RFP (5.6, 2.8, 1.4, 0.7, 0.35 µg/mL) and CaO₂ (19.14, 9.57, 4.78, 2.4, 1.2 µg/mL) at different concentrations.

Comments-5: “Why did you choose free fatty acid (lauric acid and stearic acid) at the ratio of 4:1?”

Response: Thank you for the good suggestions. In our study, we have optimized the lauric acid to stearic acid ratio based on the melting temperature. The results have been provided in the newly revised manuscript (**Please see: Table S1**) and shown in Table 2. As shown, when the ratio of lauric acid to stearic acid is 4:1, the eutectic mixture formed with a well-defined melting point at 35.2-38.3 can meet our experimental requirements, which is consistent with the results reported in previous literature (*Sol. Energy*, **2002**, **72**, **493**). So in our study, we chose the ratio of lauric acid to stearic acid at 4:1.

Table 2. The melting temperature for different ratios of lauric acid to stearic acid.

Mass ratio (LA:SA)		Melting temperature range (°C)
0.0	100.0	71.8-72.3
10.0	90.0	69.2-71.7
20.0	80.0	65.8-69.7
30.0	70.0	61.3-66.7
40.0	60.0	64.1-69.0
50.0	50.0	61.4-67.3
60.0	40.0	59.4-65.6
65.0	35.0	37.6-38.1
70.0	30.0	36.5-38.8
77.5	22.5	35.1-39.4
80.0	20.0	35.2-38.3
82.5	17.5	37.1-43.8
85.0	15.0	35.7-40.0
90.0	10.0	34.9-40.0
100.0	0.0	45.7-46.2

Comments-6: “Why did the author select rifampicin as antibiotic to treat MRSA infection? Please discuss why vancomycin which is the standard treatment for MRSA infection was not chosen.”

Response: Thank you for the good suggestions. In our study, the reasons why we chose rifampicin but not vancomycin as the antibacterial agent are as follows:

(1) RFP was primarily a frontline drug for the treatment of bacterial infection and showed good antibacterial activity in treating MRSA-related infections (*Proc. Am. Thorac. Soc.*, **2004**, **1**, **338**; *Lancet*, **2001**, **357**, **40**);

(2) In our previous research, RFP was loaded into the metal organic framework, and the combination of UV-light (365 nm), pH-triggered precise RFP release and zinc ions enables the light-activated nanocomposite to significantly inhibit MRSA-induced wound infection and accelerate wound healing (*Adv. Funct. Mater.*, **2018**, **28**, **1800011**);

(3) We obtained the size of Vancomycin and Rifampin molecules by molecular dynamic simulation. In Figure 15, the calculated data showed that RFP has a diameter of 17.96 Å and Van has a diameter of 23.53 Å. The previous experimental and theoretical work indicates most of atomic long-ranged interactions are greater than 5 Å (*Phys. Rev. Lett.*, 2004, 92, 246401), while the α -toxin pores are estimated to be 1-2.5 nm in diameter (*Biol. Cell*, 2006, 98, 667), indicating Van is more difficult to release, even via the largest pore.

Figure 15. The structure of RFP (a) and Van (b) based on the dynamic molecular simulation.

(4) Furthermore, we used vancomycin as the control, and the drug release efficiency in nanoreactors was evaluated. The results shown in Figure 16 demonstrated that the nanoreactors for loading RFP has better drug release ability, which further validated the dynamic molecular simulation data.

Figure 16 The release efficiency of nanoformulation which separately loaded Vancomycin and Rifampin.

Based on the above analyses, we chose Rifampin instead of Vancomycin as our antibacterial agent in this study.

Comments-7: *“In pore-forming assay, ANTS-DPX@PCM@Lec were synthesized. What are concentrations of ANTS and DPX used in the study and how is it fabricated?”*

Response: Thank you for the good suggestions. We synthesized the ANTS-DPX@PCM@Lec nanomaterials according to the previous research (*J. Am. Chem. Soc.*, 2011, 133, 4132; *Adv. Mater.*, 2017, 29, 1703702). In the newly revised manuscript, we have added the detailed synthesis information about the ANTS-DPX@PCM@Lec nanomaterials, including the concentrations of ANTS and DPX used in the study (**please see: P 15, Lines 12-18**).

Comments-8: *“In figure s3 (line 145-146), figure caption “The concentration of H₂O₂ generated by 1mg/mL RFP-CaO₂@PCM@Lec and RFP-CaO₂@PCM@Lec” is incorrect. This must be corrected.”*

Response: We feel sorry for this mistake in the old version. In the newly revised manuscript, we have checked the manuscript carefully, including each picture and each datum (**please see: P 9, Lines3-5**). All the modified portions are highlighted in yellow.

Comments-9: *“Axis labels and texts in several figures throughout manuscript are quite small and low resolution.”*

Response: Thanks for the good suggestion. In the newly revised manuscript, we have carefully checked the axis labels and texts for each figure, and we have also adjusted the sizes and resolution. All the modified portions are highlighted in yellow.

Comments-10: *“Nanoparticle were called differently throughout manuscript such as RFP-CaO₂@PCM@Lec, nanomachines, nanotoxoid, nano-toxoid. It is very confusing.”*

Response: Thank you for the good suggestion. In the newly revised manuscript, we have used unified terms for RFP-CaO₂@PCM@Lec, nanomachines, nanotoxoid, nano-toxoid. All the modified portions are highlighted in yellow.

Comments-11: *“If nanoparticles were claimed to neutralize toxins, hemolytic activity study should be provided.”*

Response: Thanks for the good suggestion. In this revised manuscript, we have added the hemolytic activity assay to confirm that our synthesized nanomaterials have the ability to neutralize toxins. For detailed experimental information, **please see: P 16, Lines 6-20**). The results have been provided in the newly revised manuscript (). The results are shown in Figure 17, and it can be seen that the nano-toxin (absorbed toxin) has a lower hemolysis ratio than the pure toxin.

Figure 17. Hemolysis ratio assay of different materials. (a) Representative images demonstrating the varying degrees of hemolysis; (b) Comparison of hemolysis induced by PBS, RFP-CaO₂@PCM@Lec, toxin, heated toxin and nano-toxin (n = 3; mean ± SD).

Comments-12: *“In manuscript line 109-110, it is stated that pore forming toxin was investigated by SEM in figure 2c. It was incorrect. The result in figure 2c is TEM images.”*

Response: Thank you for the good suggestion. Figure 2c shows the sample treated with phosphotungstic acid (1.5%) and then photos were taken by SEM.

“Moreover, if the pore size is 2.5 nm as mentioned in line 61, the results from TEM image in figure 2c indicates otherwise. It is inconsistency. Please discuss why did you think the arrow in figure 2c were pore-formed by toxin.”

Response: Thank you for the good suggestion. In our study, we select the toxin as a pore former which can disrupt cells by forming pores in cellular membranes and altering their permeability. The 8-aminonaphthalene-1,3,6-trisulfonic acid disodium salt (ANTS) and p-xylene-bis-pyridinium bromide (DPX) were used as a pair of fluorophore/quencher to evaluate the formation of pores, and the results showed that the presence of toxins leads to the release of the drug. To verify the relationship between toxin punching and drug release, we used a scanning electron microscope to observe the interaction between the toxin and the nanocomposite at different times. As shown in Figure 18 a-c, with the extension of treatment time, the nanoreactors structure changed significantly. With the treatment time extended to 24 h, the structure of the nanoreactors were completely collapsed. Furthermore, we used the immunogold staining to confirm the interaction between toxin and nanoreactors.

Figure 18. Representative images demonstrating the change of the nanoformulation structures. (a-c) SEM images of toxin treatments for different time periods and (d) TEM images of immunogold staining after 24 h toxin treatment.

“Please show TEM images (negative staining) of untreated nanoparticle samples as negative control to compare with treated one.”

Response: Thank you the good suggestion. In the newly revised manuscript, we have added the TEM images (negative staining) of untreated nanoparticle samples as negative control to compare with treated one (**Please see: Figure 1g**). The results are shown in Figure 19, and we can see that the untreated nanoparticle has a smooth surface and complete membrane structure; however, with the addition of toxin, the structure of the nanoparticle has changed significantly. In particular, when the treatment time is extended to 24 hours, the structure of the material has completely disintegrated (**Please see Figure 18**).

Figure 19. Representative TEM images (negative staining) of untreated nanoparticle samples.

A list of changes

Position in the Revised manuscript	Original manuscript	Revised manuscript
Page 1, Line 2	nanomachines	nanoreactors
Page 2, Line 2	nanomachines	cascade nanoreactors for
Page 2, Line 5	nanomachines	nanoreactors
Page 2, Line 6	nanomachines	nanoreactors
Page 2, Line 6	nanomachines	nanoreactors
Page 2, Line 8	nanomachines	nanoreactors, which
Page 2, Line 9	,	and
Page 3, Line 3	several intensive efforts have been made in the area of advanced functional micro- and nano-materials for delivery applications to avoid the side effects of current and developing therapies ^{3,4} .	and several intensive efforts have been made in the area of advanced functional micro- and nano-materials to avoid the side effects of current and developing therapies
Page 3, Line 5		In this case, controlled drug release systems have been developed for the purpose of maintaining a therapeutically effective drug concentration in systemic circulation for a longer period of time, as well as reducing side effects by using an active substance at the right time and place, overwhelm drug resistance mechanisms with high,

		sustained local drug concentrations
Page3, Line 9	Recently, researchers have developed self-propelled micro-/nano-machines in aqueous media, which have potential as novel and active drug delivery vehicles^{5, 6, 7}. Meanwhile, many triggering stimuli are explored for the micro-/nano-machines, including external signals, such as temperature⁸, light⁹, magnetic field¹⁰, and ultrasound¹¹, as well as physiological factors, such as pH¹², redox potential¹³, enzymes¹⁴ and biomolecules¹⁵. However, recruiting endogenous stimuli instead of external intervention for targeted delivery and controlled release represents a central goal as well as a key challenge in nanomedicine research¹⁶.	In this process, the concept of a nanoreactor was introduced for the design of a stimuli-responsive drug delivery and release nanosystem⁸⁻¹¹. The potential applications of nanoreactors are not only involved in chemical synthesis, but also in many cross-cutting fields such as biomedicine¹²⁻¹⁴. In particular, the in vivo use of micro-/nanoreactors has attracted the attention of more and more researchers for therapy and diagnosis of various diseases^{15, 16}. For construction of nanoreactors, the substrate and product should be exchanged between the inner and outer regions, that is, appropriate permeability is required for the wall of nanocompartments¹⁷. Moreover, the encapsulation of a wide variety of catalytic materials is another essential challenge. Despite the development of several nanoreactor systems, problems still remain in the encapsulation process and permeation of the substrate and products^{18, 19}.
Page4, Line 9	nothing	In recent years, a new type of functional material, phase change material (PCM), has been found to be able to quickly respond to temperature and transform

		into a transparent liquid phase for a controllable release of drugs ^{25, 26} .
Page4, Line 15	for	in
Page4, Line 15	nanomachines	nanoreactors
Page4, Line 17	nanomachines	nanoreactors
Page4, Line 17	nanomachines	nanoreactors
Page4, Line 19	nanomachines	nanoreactors
Page4, Line21	nanomachines	nanoreactors
Page4, Line 22	Hla	toxin
Page5, Line 2	nanomachines	nanoreactors
Page5, Line 3	nanomachine	nanoreactor
Page5, Line 7	nothing	It is similar to the previous report that the eutectic mixture
Page5, Line 9	exhibit	exhibits
Page5, Line 9	35.7°C -36.2°C	35.2°C -38.3°C
Page5, Line 10	nothing	(36.2°C -37.2°C) (Figure 1c and Table S1), then the DSPE-PEGylated lecithin (Lec) was used to coat the eutectic mixture and form a toxin-reactive nanoreactor for drug release, which was mixed at a mass ratio of 3:1 to prevent hemolysis and also maintain the ability to adsorb toxin. (Figure S1).

Page5, Line 13	nanomachines	nanoreactors
Page5, Line 14	Figure S1	Figure S2
Page5, Line 17	(Table S1)	(Table S2)
Page5, Line 17	The absorption peak of RFP can be detected at 473 nm when RFP-CaO ₂ @PCM@Lec was dissolved in ethanol, but not when dispersed in deionized (DI) water, then the absorption peak can be restored again with the addition of toxins.	When RFP-CaO ₂ @PCM@Lec (nanoreactors) are dissolved in ethanol, the absorption peak of RFP can be detected at 473 nm, but when dispersed in deionized (DI) water, the absorption peak cannot be detected, and then absorption peak can be recovered by adding toxin.
Page5, Line 21	5.59% and 19.14%	5.4% ± 0.9% and 17.2% ± 1.2%
Page6, Line 4	nanomachines	nanoreactors
Page6, Line 6	nanomachines	nanoreactors
Page6, Line 7	nanomachines	nanoreactors
Page6, Line 8	nanomachines	nanoreactors
Page6, Line 8	The ability of the nanomachines to capture Hla was tested by mixing the toxin with different concentrations of nanomachines and then evaluating the loading efficiency (Figure 2a). The results indicated that 100 µg of the nanomachines was sufficient to capture 4 µg of Hla, and further experimental results	100 µg of the nanoreactors was found to be able to capture 4 µg of toxin (Figure 2a and Figure S3). The immunoglod staining experiment showed that the nanoreactors without toxin treatment did not display any specific binding, while toxin-treated nanoreactor surface could combine very distinct gold nanoparticles. These results indicate that

	showed the structural integrity of these toxins was not affected by the nanomachines (Figure 2b).	nanoreactors can efficiently capture toxins without affecting their structural integrity. (Figure 2b).
Page6, Line 13	nanomachines	nanoreactors
Page6, Line 14	nanomachines	nanoreactors
Page6, Line 14	8-aminonaphthalene-1,3,6-trisulfonic acid disodium salt (ANTS) and p-xylene-bis-pyridinium bromide (DPX), which are used as a pair of fluorophore/quencher to evaluate the stability of liposomes ^{18, 26} .	The stability of liposomes was evaluated by using 8-aminonaphthalene-1,3,6-trisulfonic acid disodium salt (ANTS) and p-xylene-bis-pyridinium bromide (DPX) as a pair of fluorophore/quencher ^{21, 31} .
Page6, Line 17	nanomachines	nanoreactors
Page6, Line 21	nanomachines	nanoreactors
Page7, Line 2	nanomachines	nanoreactors
Page7, Line 3	To confirm the proposed mechanism	To confirm the above proposed mechanism,
Page7, Line 3	nanomachines	nanoreactors
Page7, Line 4	nanomachines	nanoreactors
Page7, Line 6	(35.7°C-36.2°C)	(35.2°C-38.3°C),
Page7, Line 6	nanomachines	nanoreactors
Page7, Line 7	nothing	of payloads
Page7, Line 8	nanomachines	nanoreactors

Page7, Line 10	Nothing	(Figure S4)
Page7, Line 16	Nothing	(O ₂)
Page7, Line 17	S3	S5
Page7, Line 17	RFP-CaO ₂ @PCM@Lec	nanoreactors
Page7, Line 22	RFP-CaO ₂ @PCM@Lec	nanoreactors
Page8, Line 5	RFP-CaO ₂ @PCM@Lec	the nanoreactors
Page8, Line 7	nanomachines	nanoreactors
Page8, Line 8	nanomachines	nanoreactors
Page8, Line 10	S3	S6
Page8, Line 11	RFP-CaO ₂ @PCM@Lec	nanoreactors
Page9, Line 1	nanomachines	nanoreactors.
Page9, Line 1	nanomachines	nanomachines
Page9, Line 3	3a-d	3a-3c
Page9, Line 3	nanomachines	nanoreactors
Page9, Line 7	RFP-CaO ₂ @PCM@Lec	nanoreactors
Page9, Line 9	(Figure S4)	(Figure S7).
Page9, Line 10	RFP-CaO ₂ @PCM@Lec	nanoreactors

Page9, Line 10	As shown in Figure 3e, the RFP-CaO₂@PCM@Lec almost completely inhibited the growth of bacteria, but RFP@PCM@Lec and CaO₂@PCM@Lec showed a limited antibacterial effect. The coated flat panel (Figure 3f) and live/dead staining (Figure 3g) showed similar results. Overall, RFP-CaO₂@PCM@Lec (96.71%) shows higher antibacterial activity than RFP@PCM@Lec (30.87%) and CaO₂@PCM@Lec (40.85%).	As shown in Figure 3d, nanoreactors almost completely inhibited bacterial growth, but RFP@PCM@Lec, CaO₂@PCM@Lec and PCM@Lec showed varying degrees of incomplete antibacterial effects, respectively. In vitro antibacterial activity (Figure 3e-3f) tests also showed that nanoreactors have efficient antibacterial ability (3.02 Log), and similar results were observed in live/dead staining (Figure 3g). Furthermore, we evaluated the antibacterial efficiency of RFP and CaO₂ at different concentrations, and the pure RFP and H₂O₂ (Figure S8) were shown to have limited antibacterial activity. Overall, nanoreactors (96.71%) show higher antibacterial activity than RFP@PCM@Lec (30.87%) and CaO₂@PCM@Lec (40.85%).
Page9, Line 19	Evaluation the cytotoxicity and neutralizing ability of nanomachines.	Cytotoxicity and toxin neutralizing ability of nanoreactors.

Page9, Line 19	nanomachines	nanoreactors
Page9, Line 20	nanomachines	nanoreactors
Page9, Line 22	nanomachines	nanoreactors
Page10, Line 1	nanomachines.	nanoreactors
Page10, Line 2	Histological analysis was used to evaluate whether the nanomachines caused tissue damage, inflammation or lesions. d to evaluate whether the nanomachines caused tissue damage, inflammation or lesions.	To evaluate whether the nanoreactors cause tissue damage, inflammation or lesion, histological analysis was performed in our study.
Page10, Line 4	Figure 4b	Figure S9,
Page10, Line 4	nanomachines	nanoreactor
Page10, Line7	nanomachines	nanoreactors
Page10, Line8	for	in
Page10, Line8	injury in	injured at
Page10, Line9	nothing	nanoreactors, which was further supported by
Page10, Line10	Taken together, from	Based on
Page10, Line11	nanomachines	nanoreactors
Page10, Line13	To assess the activity of toxin in the nanomachines, both terminal deoxynucleotidyl transferase dUTP nick end	Furthermore, the toxin neutralizing efficiency of nanoreactors was evaluated by measuring the hemolysis ratio of different

	labelling (TUNEL) assay and H&E assay were used to assess the toxicity of the different preparations (pure nanomachines, nanomachines detaining Hla, untreated free Hla and heat-inactivated toxins). The results are shown in Figure 4d and 4e.	nanoformulations (pure nanoreactors, free toxin, heat-inactivated toxin (heated toxin), and nanoreactors detaining toxin (nano-toxin)). As shown in Figure 4c and 4d, the untreated free toxin has high hemolytic efficiency, but after the toxin is captured by the nanoreactor, the hemolytic rate decreases significantly. Moreover, the toxicity of different nanoformulations was assessed using the terminal deoxynucleotidyl transferase dUTP nick end labelling (TUNEL) assay and the results are shown in Figure 4e and S10.
Page10, Line21	nanomachines	nanoreactors
Page10, Line22	nothing	The ability of the nanoreactors to neutralize α-toxin was further demonstrated in vivo by subcutaneous injection of pure nanoreactors, free toxin, heated toxin, and nano-toxin beneath the right flank skin of mice. Based on the skin lesions shown in Figure S11, the pure toxin induced demonstrable edema and inflammation with the extension of time (7 d, 14 d, 21 d), and this phenomenon became more and more serious, with obvious suppuration and muscle rot being observed in the skin tissue at the toxin injection site after 21 days of

		treatment. However, the nanoreactor-toxin showed no significant damage to the skin. Furthermore, the H&E, immunocytochemistry (IHC) and blood routine assays were used to evaluate the toxicity of different nanoformulations at 21 days post injection. The toxin treatment was shown to induce stronger tissue damage, inflammation or lesion by H&E and IHC analysis (Figure 4f), in contrast to a similar result between the nanoreactor-toxin and the control, which was further supported by the analysis results of blood routine (Figure S12). All the above test results reveal that the nanoreactors can effectively neutralize toxins without causing significant cytotoxicity or physiological toxicity.
Page11, Line13	nanomachines	nanoreactors
Page13, Line19	RFP-CaO ₂ @PCM@Lec	nanoreactors
Page13, Line19	Figure 6a	Figure 5a
Page12, Line1	Figure 5b	Figure 5b
Page12, Line2	RFP-CaO ₂ @PCM@Lec	nanoreactors
Page12, Line3	nanomachine	nanoreactor

Page12, Line5	Figure 6c	(Figure 5c).
Page12, Line7	RFP-CaO ₂ @PCM@Lec	nanoreactor
Page12, Line8	Figure 6c and 6e	(Figure 5d and 5e)
Page12, Line10	nothing	(Figure S13), and the existence of nanoreactors was shown to effectively remove bacteria from various organs. The nanoreactor treatment significantly
Page12, Line12	good	better
Page12, Line12	relative to	than
Page12, Line14	Figure 6f and 6g	Figure 5f and 5g
Page12, Line16	RFP-CaO ₂ @PCM@Lec	nanoreactors.
Page12, Line17	(Figure 6g)	(Figure 5g)
Page12, Line19	RFP-CaO ₂ @PCM@Lec	nanoreactor
Page12, Line21	Evaluation of immune effect of nanomachines. Following the safety assessment, we studied the ability of the nanomachines to elicit potent humoral immunity (Figure 5a). The induction of germinal centers in lymph nodes is one of the key steps in the immune response to infection, and affinity-based maturation of B cells occurs in these regions³⁶.	Immune effect and in vivo toxin neutralization of nanoreactors. Following the in vivo antibacterial assessment, we studied the ability of the nanoreactors to elicit potent humoral immunity (Figure 6a). Germinal centers (GCs) are the primary sites for the affinity-based maturation of B cells, with the affinity of serum antibodies increasing with time after immunization^{42, 43}. These

		high-affinity antibodies of specific isotypes provide excellent protection against a variety of pathogenic microbial infections.
Page13, Line3	effect	performance
Page13, Line4	nothing	immune effect and in vivo toxin neutralization
Page13 Line5	21 d	at 21 days post
Page13 Line7	nanomachine	nanoreactors
Page13, Line8	(Figure 5b)	Figure 6b and Figure S14
Page13, Line10	amd	and
Page13, Line10	nothing	enzyme-linked immunosorbent assay (ELISA)
Page13, Line12	(Figure 5c)	(Figure 6c).
Page13, Line12	nano-toxoid	nano-toxin
Page13, Line13	Our study showed that the nanotoxoid	In the present study, the nano-toxin
Page13, Line14	compared to	than
Page13, Line15	nothing	Furthermore, the in vivo toxin neutralization ability of nanoreactors was evaluated by measuring hemolysis ratio (Figure 6d and 6e). It can be seen that the nanoreactors have better toxin-neutralizing ability and can significantly reduce the hemolysis rate. Finally, the protective immunity

		bestowed by the nanoreactors was evaluated by subjecting the vaccinated mice to toxin administration at a toxin dose of 120 µg/kg ⁴⁵ , which resulted in 100% mortality within 2 h in the unvaccinated group. Meanwhile, the nano-toxin boosters improved the survival rate to 100% versus an 80% survival rate for the heat-treated toxin vaccine with boosters (n=10) (Figure 6f).
Page14, Line3	nanoparticle	nanoreactors
Page14, Line4	nanomachines	nanoreactors
Page14, Line7	nanomachines	nanoreactors
Page14, Line8	nanomachines	nanoreactors
Page14, Line9	nanomachines	nanoreactors
Page14, Line11	nanomachines	nanoreactors
Page14, Line12	nano-machines	nanoreactor-toxin
Page14, Line13	therapy	therapeutic
Page15, Line2	nanoparticles.	nanoreators.
Page15, Line10	nanoparticles	nanoreactors
Page15, Line12	nothing	Synthesis of ANTS-DPX@PCM@Lec. The phospholipid solution (15 mL) was heated to 50°C.

		The PCM solution (3 mL) was mixed with the desired payloads (500 μL 12.5 mM of ANTS and 500 μL 45 mM of DPX in DMSO)21, 26 and then added dropwise into the preheated phospholipid solution, followed by vigorous vortex for 2 min. After cooling in ice water for 60 min, the cloudy solution was centrifuged for removing the un-encapsulated molecules and then filtered through a 0.22 micron filter. After washing three times with water, the resultant nanoreactors were suspended in water at 4°C for further use.
Page16, Line3	RFP-CaO ₂ @PCM@Lec was	nanoreactors were
Page16, Line6	nothing	Evaluation of toxin adsorption and hemolysis of nonareactors by using lecithin and DSPE-PEG nanomaterials at different ratios. BCA Protein Assay Kit was used for quantitative detection of the adsorption of toxins by materials. Briefly, 200 μL of 500 μg/mL nonareactors synthesized in different mass proportions (Lec : DSPE-PEG=1:1,3:1,6:1,9:1, 12:1 and 1:0) was mixed with 10 μL of 400 ug/mL toxin to interact with each other at 37°C for two hours, using PBS as a control. The mass of the adsorbed toxin

		was calculated by the absorbance at 462 nm according to the detection method of the BCA kit. Under the same experimental protocol, the hemolysis rate of the material can also be calculated by the following formula. Briefly, 150 μL of different materials synthesized at different proportions (Lec: DSPE-PEG=1:1,3:1,6:1,9:1, 12:1 and 1:0) and 150 μL of 2% RBCs were incubated for 30 min at room temperature. After centrifugation at 2 000 x g for 5 min, the hemolysis was determined for each sample by measuring the absorbance of the supernatant at 540 nm using a microplate reader. A 100% lysis control was prepared by treating RBCs with Triton X-100. The hemolysis rate of each group was calculated as follows. $\text{Hemolysis rate} = \frac{\text{Abs}(\text{experiment})}{\text{Abs}(X - 100)} \times 100\% (1)$
Page16, Line21	nothing	Bacterial culture. Briefly, 200 μL of 10⁸ CFU/mL bacteria was incubated with different concentrations of nanoreactors, RFP and CaO₂ at 37°C for 2 h at 120 rpm. To evaluate the bacterial mortality, the treated bacteria were diluted and uniformly

		plated in Luria-Bertani (LB) solid medium, followed by incubation at 37 °C for 24 h. Finally, colony forming unit (CFU) was counted and compared with the control plate. Each treatment was prepared in triplicate and the mean values were compared with each other.
Page17, Line4	nothing	In Vivo Safety. Briefly, the BALB/c mice (6-8 weeks old) were first shaved to remove the hair on the back. Subsequently, 200 µL of 100 µg/mL of nanoreactors (20 µg) was injected subcutaneously, using PBS as a control. At 24 h post injection, the mice were euthanized, and the internal organs (heart, liver, spleen, lung, kidney) were collected for histological analysis by hematoxylin and eosin (H&E) staining. Meanwhile, the plasma was collected for biochemical indicator detection (ALB, ALP, ALT, AST, A/G, BUN, GLOB, TP). Assessment was also performed on the toxicity of nanoreactors (100 µg), toxin (4 µg), heated toxin (4 µg, 70°C inactivated for 1h) and nano-toxin (4 µg toxin absorbed by 100 µg RFP-CaO₂@PCM@Lec) using PBS as control. Briefly, BALB/c mice were first shaved to remove the hair on their back and the

		above materials were injected subcutaneously and separately to each group of mice. At 24 h post injection, the mice were euthanized, and skin samples at the injection site were collected for histological analysis by hematoxylin and eosin (H&E) and TUNEL. TUNEL staining and Ipwin32 software were used to count the number of cells with a different color fluorescence. After 21 days of immunization, hematoxylin and eosin (H&E) skin staining and immunohistochemistry (IHC) were performed on the dorsal skin of each group to judge the viable toxicity of different treatments. At the same time, the blood of the mice was collected, and blood routine tests were performed to observe the number of white blood cells (WBC) and neutrophils (Gran).
Page18, Line1	nothing	Inhibitory effect of nanoformulations on hemolysis. The ability of nanoformulations to prevent hemolysis was investigated under five different experimental groups: PBS, nanoreactors (100 µg), toxin (4 µg), heated toxin (4 µg ,70°C inactivated for 1h) and nano-toxin (4 µg toxin absorbed by 100 µg nanoreactors). Briefly, 150

		 μL of different materials and 150 μL of 2% red blood cells (RBCs) were incubated for 30 min at room temperature, followed by centrifugation at 2 000 x g for 5 min. Next, the hemolysis of each group was determined by measuring the absorbance of the supernatant at 540 nm using a microplate reader. Meanwhile, a 100% lysis control was prepared by treating RBC with Triton X-100. Finally, the hemolysis rate of each group was calculated according to formula 1. 
Page18, Line20	RFP-CaO ₂ @PCM@Lec	nanoreactors
Page19, Line 1	nothing	 All animal experiments were in compliance with the Huazhong Agriculture University (HZAUMO-2018-036 , approved by The Scientific Ethic Committee of Huazhong Agricultural University) 
Page19, Line13	RFP-CaO ₂ @PCM@Lec(1 00 μ g), toxin(4 μ g), heated toxin(4 μ g) and adsorbed toxin by	nanoreactors (100 μ g), toxin (4 μ g), heated toxin (4 μ g ,70°C inactivated for 1h) and nano-toxin (4 μ g toxin

	RFP-CaO ₂ @PCM@Lec (100µg RFP-CaO ₂ @PCM@Lec incubated with 4µg toxin),	absorbed by 100 µg nanoreactors)
Page19, Line16	For flow cytometric analysis, the lymph nodes were dissociated into single cell suspensions.	the lymph nodes were collected and dissociated into single cell suspensions for flow cytometric analysis
Page19, Line17	nothing	After staining
Page19, Line19	Analysis was performed using Flowjo software.	data were collected on a flow cytometer and analyzed using Flowjo software.
Page19, Line21	RFP-CaO ₂ @PCM@Lec(100µg), toxin(4µg), heated toxin(4µg) and adsorbed toxin by RFP-CaO ₂ @PCM@Lec (100µg RFP-CaO ₂ @PCM@Lec incubated with 4µg toxin)	nanoreactors (100 µg), toxin (4 µg), heated toxin (4 µg ,70°C inactivated for 1h) and nano-toxin (4 µg toxin absorbed by 100 µg nanoreactors)
Page20, Line 1	assay	measuring
Page20, Line3	nothing	Next
Page20, Line4	nothing	followed by the addition
Page20, Line6	finished the reaction	was terminated by adding
Page20, Line7	measured at 450 nm with a Microplate reader.	Finally, toxin-specific antibody titers were measured at 450 nm using a Microplate reader.
Page20, Line9	nothing	Toxin neutralizing ability of nanoreactors. After 21 days of immunization, sera were collected from the different experimental groups of mice. Briefly, 20 µL serum was incubated with 10 µL of 50 µg/mL toxin and 20 µL of

		Hank solution for 30 min at room temperature, followed by the addition of 50 μ L of 2% RBCs and incubation for another 30 min. According to the above experimental method, the hemolysis efficiency after serum toxin neutralization can be calculated by formula 1.
Page20, Line15	nothing	Mouse survival rate. After the end of the 21-day immunization, the mice treated with PBS, nanoreactors (100 μ g), toxin(4 μ g), heated toxin (4 μ g ,70°C inactivated for 1h) and nano-toxin (4 μ g toxin absorbed by 100 μ g nanoreactors) were injected in the tail vein with toxin at the dosage of 120 μ g/kg, and the survival rate of the decimals was observed for each group within 360 hours.
Page20, Line20	nothing	Statistical analysis. All the results were presented as mean \pm SEM from at least three independent experiments. Statistical analyses were performed using the t-test. Values of *p < 0.05, **p < 0.01 and ***p < 0.001 were considered statistically significant.
Page21, Line 1	Supporting Information. Additional details and images related to the characterization of RFP-CaO ₂ @PCM@Lec nanoparticles; UV	Supporting Information. Additional details and images related to the characterization of nanoreactors; The different mass ratios for lecithin to

	absorption spectra of RFP; H₂O₂ generated by RFP-CaO₂@PCM@Lec; Coated flat panel of bacterium incubated with different concentration of RFP-CaO₂@PCM@Lec; Inhibition rate of bacterium incubated with different concentration of RFP-CaO₂@PCM@Lec; Live/dead staining of bacterium incubated with different concentration of RFP-CaO₂@PCM@Lec; Toxicity of toxins to B. Subtilis; The antibacterial efficiency against wound; The number of bacteria in the organs; Characterization of CaO₂.	DSPE-PEG; TEM and SEM image of the nanoreactors; Detection of the adsorption of toxins by different quality materials using BCA Protein Assay Kit; The structure of RFP at the level of 6-311G by using ORCA program; H₂O₂ generated by nanoreactors; UV absorption spectra of RFP; Toxicity of toxins to B. Subtilis; The antibacterial activity of different nanomaterials; Histological analysis of internal organs; The quantitative analysis of TUNEL staining; In vivo toxin neutralization; the blood routine analysis; The number of bacteria in the organs; The flow cytometric data of different treatment; Characterization of CaO₂; The melting temperature of different ratio LA to SA; The loading rate of RFP and CaO₂ for different materials.
Page23, Line3	nanomachine	RFP-CaO ₂ @PCM@Lec nanoreactors
Page23, Line5	(c) Differential scanning calorimetry (DSC) curves of LA and SA at a weight ratio of 4:1	(c) Differential scanning calorimetry (DSC) curves of LA and SA. (d) A typical TEM image of the RFP-CaO ₂ @PCM@Lec nanoreactors.
Page23, Line9	RFP-CaO ₂ @PCM@Lec nanoparticles.	RFP-CaO ₂ @PCM@Lec nanoreactors.
Page24, Line2	nanomachine	RFP-CaO ₂ @PCM@Lec nanoreactors

Page24, Line3	nanotoxoid	nano-toxin
Page24, Line7	RFP-CaO ₂ @PCM@Lec	RFP-CaO ₂ @PCM@Lec nanoreactors
Page24, Line11	The size changes after toxin was anchored into nanomachines;	(i) The size changes after toxin was anchored into RFP-CaO ₂ @PCM@Lec nanoreactors
Page24, Line13	RFP-CaO ₂ @PCM@Lec	RFP-CaO ₂ @PCM@Lec nanoreactors
Page25, Line5	RFP-CaO ₂ @PCM@Lec	RFP-CaO ₂ @PCM@Lec nanoreactors
Page25, Line8	RFP-CaO ₂ @PCM@Lec	RFP-CaO ₂ @PCM@Lec nanoreactors
Page25, Line10	(e) bacterial inhibition rate of MRSA incubated with 100 µg/mL of different materials at 37°C for 2 h; (f) Coated flat panel and	(e) bacterial inhibition rate of MRSA incubated with 100 µg/mL of different materials at 37°C for 2 h; (f) Coated flat panel and
Page25, Line13	RFP-CaO ₂ @PCM@Lec	RFP-CaO ₂ @PCM@Lec nanoreactors
Page26, Line2	RFP-CaO ₂ @PCM@Lec	RFP-CaO ₂ @PCM@Lec nanoreactors
Page26, Line3	RFP-CaO ₂ @PCM@Lec	RFP-CaO ₂ @PCM@Lec nanoreactors
Page26, Line5	RFP-CaO ₂ @PCM@Lec	RFP-CaO ₂ @PCM@Lec nanoreactors
Page26, Line8	RFP-CaO ₂ @PCM@Lec	RFP-CaO ₂ @PCM@Lec nanoreactors
Page26, Line9	nothing	Immunocytochemistry (IHC) assays were used to evaluate the toxicity of different nanoformulations at 21 days post injection.
Page28, Line 2	Figure 5. (a)The scheme of nanomachines stimulate the body's immune response and improve the	Figure 6. (a)The scheme demonstrating that nanoreactors stimulate the body's immune response and

	therapy effect of bacterial infected mice. (b) Flow cytometric analysis of cells at the draining lymph node at 21 d after administration with blank nanomachines, toxin, heated toxin, or nanomachine-captured toxin (n = 6; mean ± SD). Cells were first gated on the B220+IgDlow population and values are expressed as percentage GL-7+; (c) Multivalent antibody responses in vivo. Mice were vaccinated with blank solution, heat-treated hSP, or nanotoxoid (hSP) on day 0 with boosts on day 7 and 14. On day 21, the serum was sampled and analyzed for the presence of IgG antibody titers against toxin.	improve the therapeutic effect of bacterially infected mice. (b) Flow cytometric analysis of cells in the draining lymph node at 21 days post administration with RFP-CaO₂@PCM@Lec nanoreactors, toxin, heated toxin, or nano-toxin (n = 6; mean ± SD). Cells were first gated on the B220+IgDlow population and values are expressed as percentage GL-7+; (c) Multivalent antibody responses in vivo. Mice were vaccinated with RFP-CaO₂@PCM@Lec nanoreactors, toxin, heated toxin, or nano-toxin on day 0 with boosts on day 7 and 14. On day 21 post first vaccination, the serum was sampled and analyzed for the presence of IgG antibody titers against toxin. (d) Representative images demonstrating the varying degrees of hemolysis; (e) Comparison of hemolysis induced by antibody generated by injection of PBS, RFP-CaO₂@PCM@Lec nanoreactors, toxin, heated toxin and nano-toxin (n = 3; mean ± SD).(f) Survival rates of mice over 15 days following an intravenous injection of α-toxin (120 µg/kg).
--	---	---

Reviewers' comments:

Reviewer #1 (Remarks to the Author):

The authors have vastly improved the content and scholarly presentation in the revised manuscript, and the data and narrative in the present version is of sufficient quality for publication in Nature Communications. Some minor suggestions remain, however, and should be addressed.

1. The authors mention the use of "DSPE-PEGylated lecithin" to prepare the nanoreactor. The term is confusing and I believe the authors mean DSPE-PEG and lecithin as two separate entities. This should be corrected.
2. Data between figure 4d and figure 6d do not match. In figure 4 d, the nanoreactor/toxin complex is shown to have residual hemolytic activity but in figure 6d, no hemolysis was observed. The inconsistency should be corrected.
3. Figure 6b has a high overlap of error bars, which makes the reported statistical significance hard to believe. The authors should double check the statistical analysis and data presentation.

Reviewer #2 (Remarks to the Author):

Authors have revised the paper carefully following the comments of the reviewers point by point.

It can be published as is.

Point-by-point Response

Reviewer #1

Thanks for the positive comment and constructive suggestions on how to improve our manuscript.

Comments-1: *“The authors mention the use of "DSPE-PEGylated lecithin" to prepare the nanoreactor. The term is confusing and I believe the authors mean DSPE-PEG and lecithin as two separate entities. This should be corrected.”*

Response: Thank you for the good suggestion. In our study, Lecithin and DSPE-PEG3400 were used to coat the eutectic mixture of two fatty acids as a gate material in fabricating toxin-responsive nanoreactors for drug release. DSPE-PEG and lecithin as two separate entities. In the newly revised manuscripts, we have corrected the “DSPE-PEGylated lecithin”, the modified portions are highlighted in yellow (Please see: P4, L14).

Comments-2: *“Data between figure 4d and figure 6d do not match. In figure 4 d, the nanoreactor/toxin complex is shown to have residual hemolytic activity but in figure 6d, no hemolysis was observed. The inconsistency should be corrected.”*

Response: We are very sorry, probably because we are not clear enough in the article, which leads to the reviewer's misunderstanding of figure 4d and figure 6d.

In the figure 4d, we have evaluated the hemolysis induced by PBS, RFP-CaO₂@PCM@Lec nanoreactors, toxin, heated toxin and nanoreactor/toxin (the detailed experimental methods, please see: P16, L6-20), it can be seen that the nanoreactor/toxin have lower hemolysis ratio compare with the pure toxin when these materials were directly incubated with red blood cells (RBCs), these results revealed that the toxin were effectively captured by the nanoreactors so that significantly

inhibiting hemolysis. However, in the figure 6d, it display the hemolysis ratio assay of antibody induced by different materials (the detailed experimental methods, Please see: P 20, L 8-13), it can be seen that when the toxin were captured by nanoreactors which have better toxin-neutralizing ability and can significantly reduce the hemolysis rate, thereby the figure 6d the nanoreactor/toxin complex have no hemolysis was observed.

Comments-3: *“Figure 6b has a high overlap of error bars, which makes the reported statistical significance hard to believe. The authors should double check the statistical analysis and data presentation.”*

Response: Thank you for the good suggestion. In the newly revised manuscripts, we have added the double check the statistical analysis and provided the data (Please see: P13, L8). These results showed that the toxin captured by the nanoreactors significantly increased the percentage of germinal center labeled GL-7 B cells compared with the toxin group (P=0.04).

Reviewer #2

Thanks for the positive comment and constructive suggestions on how to improve our manuscript.

A list of changes

Position in the Revised manuscript	Original manuscript	Revised manuscript
---	----------------------------	---------------------------

P1, L7-8	nothing	No. 1 Shizishan Street, Hongshan District, Wuhan, Hubei
P4, L13	Figure 1a	Fig. 1a
P4, L14	DSPE-PEGylated-lecithin (Lec) is	lecithin (Lec) and DSPE- PEG3400 are
P4, L15	Figure 1b	Fig. 1b
P5, L10	Figure 1c	Fig. 1c
P5, L10	Table S1	Supplementary Table S1
P5, L10	DSPE-PEGylated-lecithin (Lec) was	Lec and DSPE-PEG3400 were
P5, L13	Figure S1	Supplementary Figure S1
P5, L14	Figure 1d	Fig. 1d
P5, L15	Figure S2	Supplementary Figure S2
P5, L17	Table S2	Supplementary Table S2
P5, L17	Figure 1e	Fig. 1e
P5, L22	Figure 1f	Fig. 1f
P6, L5	Figure 1g	Fig. 1g
P6, L6	Endogenous stimulus- triggered drug release from the nanoreactors.	Stimulus triggered drug release from the nanoreactors.
P6, L8	Figure 2a	Fig. 2a
P6, L9	Figure S3	Supplementary Figure S3
P6, L12	Figure 2b	Fig. 2b
P6, L14	Figure 2c	Fig. 2c
P6, L16	Figure 2d	Fig. 2d
P7, L6	Figure 2e	Fig. 2e
P7, L10	Figure S4	Supplementary Fig. S4
P7, L15	Figure 2f	Fig. 2f

P7, L15	Figure 2g	Fig. 2g
P7, L17	Figure 2g	Fig. 2g
P7, L17	Figure S5	Supplementary Fig. S5
P8, L4	Figure 2h	Fig. 2h
P8, L9	Figure 2i	Fig. 2i
P8, L11	Figure S6	Supplementary Fig. S6
P8, L11	Figure 2j	Fig. 2j
P8, L16	Figure 2j	Fig. 2j
P9, L3	Figure 3a-3c	Fig. 3a-3c
P9, L5	$\mu\text{g} / \text{mL}$	$\mu\text{g mL}^{-1}$
P9, L7	$\mu\text{g} / \text{mL}$	$\mu\text{g mL}^{-1}$
P9, L9	Figure S7	Supplementary Fig. S7
P9, L10	$\mu\text{g} / \text{mL}$	$\mu\text{g mL}^{-1}$
P9, L10	Figure 3d	Fig. 3d
P9, L13	Figure 3e-3f	Fig. 3e-3f
P9, L15	Figure 3g	Fig. 3g
P9, L16	Figure S8	Supplementary Fig. S8
P10, L1	$\mu\text{g} / \text{mL}$	$\mu\text{g mL}^{-1}$
P10, L2	Figure 4a	Fig. 4a
P10, L4	Figure S9	Supplementary Fig. S9
P10, L10	Figure 4b	Fig. 4b
P10, L15	Figure 4c and 4d	Fig. 4c and 4d
P10, L19	Figure 4e	Fig. 4e
P10, L19	Figure S10	Supplementary Fig. S10
P11, L3	Figure S11	Supplementary Fig. S11
P11, L8	Figure 4f	Fig. 4f
P11, L12	Figure S12	Supplementary Fig. S12
P11, L19	Figure 5a	Fig. 5a
P11, L21	CFU / mL	CFU mL ⁻¹

P12, L1	Figure 5b	Fig. 5b
P12, L5	Figure 5c	Fig. 5c
P12, L8	Figure 5d and 5e	Fig. 5d and 5e
P12, L12	Figure S13	Supplementary Fig. S13
P12, L14	Figure 5f and 5g	Fig. 5f and 5g
P12, L17	Figure 5g	Fig. 5g
P12, L23	Figure 6a	Fig. 6a
P13, L1	Immune effect and in vivo toxin neutralization of nanoreactors.	Immunity and in vivo toxin neutralization of nanoreactors
P13, L8	nothing	P=0.04
P13, L8	Figure 6b	Fig. 6b
P13, L8	Figure S14	Supplementary Fig. S14
P13, L13	Figure 6c	Fig. 6c
P13, L16	Figure 6d and 6e	Fig. 6d and 6e
P13, L20	$\mu\text{g} / \text{kg}$	$\mu\text{g kg}^{-1}$
P13, L22	Figure 6f	Fig. 6f
P15, L2	Fabrication of liposome-encapsulated phase change material (PCM) nanoreactors	Fabrication of nanoreactors
P15, L3	mg / mL	mg mL^{-1}
P15, L4	mg / mL	mg mL^{-1}
P15, L6	mg / mL	mg mL^{-1}
P16, L7	$\mu\text{g} / \text{mL}$	$\mu\text{g mL}^{-1}$
P16, L9	$\mu\text{g} / \text{mL}$	$\mu\text{g mL}^{-1}$
P17, L4	$\mu\text{g} / \text{mL}$	$\mu\text{g mL}^{-1}$
P18, L13	CFU / mL	CFU mL ⁻¹
P18, L19	mg / mL	mg mL^{-1}

P20, L17	$\mu\text{g} / \text{kg}$	$\mu\text{g kg}^{-1}$
P21, L17	The manuscript was written through contributions of all authors. All authors have given approval to the final version of the manuscript.	Y. W, Z. Y. S and H. J. W performed experiments. Y. W and Z. Y. S data analysis. Y. W, Z. Y. S and H. Y. H designed experiments, interpreted results and wrote the manuscript.
P22, L8	nothing	Data availability The authors declare that data supporting the findings of this study are available within the paper and its supplementary information files.